# FEDMUON: FEDERATED LEARNING WITH BIAS-CORRECTED LMO-BASED OPTIMIZATION

**Yuki Takezawa**
Kyoto University, OIST

**Anastasia Koloskova**
University of Zurich

**Xiaowen Jiang**
CISPA, Saarland University

**Sebastian U. Stich**
CISPA

## ABSTRACT

Recently, a new optimization method based on the linear minimization oracle (LMO), called Muon, has been attracting increasing attention since it can train neural networks faster than existing adaptive optimization methods, such as Adam. In this paper, we study how Muon can be utilized in federated learning. We first show that straightforwardly using Muon as the local optimizer of FedAvg does not converge to the stationary point since the LMO is a biased operator. We then propose FEDMUON that can mitigate this issue. We also analyze how solving the LMO approximately affects the convergence rate and find that, surprisingly, FEDMUON can converge for any number of Newton-Schulz iterations, while it can converge faster as we solve the LMO more accurately. Through experiments, we demonstrated that FEDMUON can outperform the state-of-the-art federated learning methods.

## 1 INTRODUCTION

Federated learning, which can train neural networks in parallel across many clients, has been attracting much attention (Kairouz et al., 2021; McMahan et al., 2017; Karimireddy et al., 2020). In federated learning, each client has its own training datasets and updates its parameters using a local optimizer, such as SGD. The central server collects the parameters from the clients and aggregates them. Since clients do not need to share their local training datasets with others, federated learning inherently preserves data privacy.

For training neural networks efficiently, using an appropriate stepsize is one of the most critical factors. If the stepsize is too large, the training collapses, whereas if the stepsize is too small, the training requires a huge number of iterations. To adjust the stepsize on the fly during the training, using adaptive optimization methods, such as AdaGrad (Duchi et al., 2011), Adam (Kingma & Ba, 2017), Shampoo (Gupta et al., 2018), and other methods (Loshchilov & Hutter, 2019; Vyas et al., 2025), have long been regarded as the de facto standard for training neural networks.

Recently, Muon (Liu et al., 2025a) has emerged as a promising alternative, attracting significant attention. Many papers evaluated the performance of Muon and demonstrated that Muon can train neural networks faster and achieve higher accuracy than the existing optimization methods, such as AdamW (Liu et al., 2025a; Semenov et al., 2025). Roughly speaking, Muon projects the momentum in the Momentum SGD onto the space of orthogonal matrices. Muon is closely related to various optimization methods: it can be interpreted as a simplified version of Shampoo (Gupta et al., 2018), in which a certain momentum accumulation is disabled (Liu et al., 2025a), and as an instance of optimizers with linear minimization oracle (LMO) under a specific norm (Pethick et al., 2025). Kovalev (2025) also showed that Muon is a special instance of the trust-region optimization method.

To use Muon for the large-scale training, developing the distributed version of Muon is important. However, Muon requires us to solve the LMO every iteration, which makes it difficult to straightforwardly use Muon in a distributed environment. Ahn et al. (2025) proposed a method to solve the LMO in a distributed manner, although their method does not support multiple local steps and incurs a huge communication cost. Thérien et al. (2025) proposed MuLoCo, which extends Muon

by allowing clients to update the parameters multiple times by Muon as in Local SGD (Stich, 2019; Woodworth et al., 2020). Although Thérien et al. (2025) demonstrated that MuLoCo performs well when all clients share the same training dataset, their method is limited to homogeneous settings and lacks theoretical guarantees. As we show in Section 3, MuLoCo fails to converge when clients have different datasets, which is a fundamental characteristic of federated learning.

In this paper, we study the federated learning methods with the LMO and propose FEDMUON. (i) First, we show that straightforwardly using Muon as the local optimizer in FedAvg failed to converge to the stationary point since the LMO is a biased operator. We formally analyze the lower bound of this straightforward method, showing that it does not converge to the stationary point, especially in the heterogeneous setting. (ii) We then propose FEDMUON, which can mitigate the bias caused by the LMO and can provably converge to the stationary point. (iii) Furthermore, we derive a novel analysis and reveal how the inexact LMO affects the convergence behavior of FEDMUON. Since solving the LMO exactly is computationally expensive, we solve the LMO approximately by running the Newton-Schulz iteration (Schulz, 1933) several times in practice. There were many papers that analyzed the convergence behavior of Muon (Riabinin et al., 2025; Liu et al., 2025a; Shen et al., 2025), while most of them assumed that the LMO is solved exactly and ignored the effect caused by the inexact LMO. We analyze the impact of inexact solutions to the LMO on the convergence rate. We discover that for any number of Newton-Schulz iterations, FEDMUON can converge to the stationary point and can converge faster by up to a factor proportional to the square root of the dimension of the parameters as we solve the LMO more accurately. We experimentally demonstrated the effectiveness of FEDMUON, showing that FEDMUON can achieve higher accuracy than the state-of-the-art adaptive federated learning optimization methods.

Our contributions are summarized as follows:

- We show that directly plugging Muon into FedAvg as the local optimizer does not converge to the stationary point since the LMO is a biased operator.
- We propose FEDMUON, which mitigates the above issue by the bias correction mechanism and can converge to the stationary point.
- We analyze the convergence rate of FEDMUON with the inexact LMO. Then, we show that for any number of Newton-Schulz iterations, FEDMUON can converge, revealing how the Newton-Schulz iteration affects the convergence rate.
- Through the experiments, we demonstrated that FEDMUON can outperform the state-of-the-art federated learning methods.

**Notation:** We use $\|\cdot\|$ to denote an arbitrary norm, and its dual norm is denoted by $\|\cdot\|_\star$. When we refer to a specific norm, we explicitly use the notation $\|\cdot\|_p$, $\|\cdot\|_F$, $\|\cdot\|_{sp}$, and $\|\cdot\|_{trace}$ to denote the Schatten $p$-norm, Frobenius norm, spectral norm, and trace norm, respectively. We denote $[n] = \{1, 2, \ldots, n\}$ for any $n \in \mathbb{N}$.

## 2 PRELIMINARY

In this section, we briefly introduce federated learning and Muon. The detailed discussion about the related works is deferred to Appendix B.

**Federated Learning:** We consider the following problem where the loss functions are distributed among $n$ clients:

$$\min_{\boldsymbol{X} \in \mathcal{X}} \left[ f(\boldsymbol{X}) \coloneqq \frac{1}{n} \sum_{i=1}^{n} f_i(\boldsymbol{X}) \right], \qquad f_i(\boldsymbol{X}) \coloneqq \mathbb{E}_{\xi_i \sim \mathcal{D}_i}[F_i(\boldsymbol{X}; \xi_i)],$$

where $\mathcal{X}$ is the parameter space (e.g., $\mathbb{R}^d$ or $\mathbb{R}^{d_1 \times d_2}$), $\boldsymbol{X}$ is the model parameter, $\mathcal{D}_i$ is the training that client $i$ holds, and $f_i : \mathcal{X} \to \mathbb{R}$ is the loss function of client $i$.

The most fundamental algorithm for federated learning is Federated Averaging (FedAvg) (McMahan et al., 2017). In FedAvg, each client updates the parameter by using its own loss function, and then the central server aggregates the parameters sent from the clients. The update rule of FedAvg is described in Appendix C. The original FedAvg uses SGD as the local optimizer, while, as in the non-distributed learning, it is important to use adaptive optimization methods for stable and fast training. Many papers proposed federated learning methods that use more sophisticated optimizers,

such as Momentum SGD (Lin et al., 2021), Adam (Reddi et al., 2021), and the Newton method (Elgabli et al., 2022). Reddi et al. (2021) proposed a general framework and analyzed the convergence rate with various optimizers.

**Optimizer with Linear Minimization Oracle:** Recently, optimizers with linear minimization oracle (LMO) have been attracting a lot of attention (Liu et al., 2025a; Pethick et al., 2025; Riabinin et al., 2025). LMO is defined as follows:

$$\text{lmo}(\boldsymbol{X}; \mathcal{D}) := \arg\min_{\boldsymbol{Y} \in \mathcal{D}} \langle \boldsymbol{X}, \boldsymbol{Y} \rangle,$$

where $\mathcal{D}$ is the convex set and $\langle \boldsymbol{X}, \boldsymbol{Y} \rangle := \sum_{i,j} X_{ij} Y_{ij}$. Originally, the LMO has been used to solve the convex constrained problems in the Frank-Wolfe algorithm (Frank & Wolfe, 1956; Jaggi, 2013). Recently, Jordan et al. (2024) proposed Muon, which uses LMO for training neural networks, which is an unconstrained optimization problem. They showed that Muon can train neural networks faster than AdamW (Loshchilov & Hutter, 2019) and Shampoo (Gupta et al., 2018; Shi et al., 2023), which are the most commonly used optimizers these days.

Specifically, the optimizers with LMO choose the unit ball as the constraint set $\mathcal{D}$, measured in any chosen norm $\|\cdot\|$. With a slight abuse of notation, we use $\text{lmo}(\cdot)$ to represent $\text{lmo}(\cdot; \mathcal{D})$ with $\mathcal{D} := \{\boldsymbol{Y} \in \mathcal{X} \mid \|\boldsymbol{Y}\| \leq 1\}$, i.e.,

$$\text{lmo}(\boldsymbol{X}) := \arg\min_{\boldsymbol{Y} \in \{\boldsymbol{Y} \in \mathcal{X} \mid \|\boldsymbol{Y}\| \leq 1\}} \langle \boldsymbol{X}, \boldsymbol{Y} \rangle.$$

Then, the update rules are given by:

$$\boldsymbol{M}^{(r+1)} = (1 - \alpha)\boldsymbol{M}^{(r)} + \alpha \nabla F(\boldsymbol{X}^{(r)}; \xi^{(r)}),$$

$$\boldsymbol{X}^{(r+1)} = \boldsymbol{X}^{(r)} + \eta \text{lmo}(\boldsymbol{M}^{(r+1))}).$$

By varying the norm, we can recover different popular optimizers. Specifically, if parameter space is a vector when we choose the Euclidean norm and max norm, we can recover Normalized SGD with momentum (Cutkosky & Mehta, 2020) and Sign SGD with momentum (Sun et al., 2023), respectively. Then, if the parameter space is $\mathbb{R}^{d_1 \times d_2}$ and we use the spectral norm for the LMO, we can obtain Muon (Jordan et al., 2024). Note that the parameter space needs to be the space of $d_1 \times d_2$ matrices for Muon. Each layer is taken into account separately. For instance, the parameter of the convolutional layer is *out_channel* $\times$ *in_channel* $\times h \times w$ matrix. When we use Muon, we consider $d_1 = out\_channel$ and $d_2 = in\_channel \times h \times w$. The remaining scalar and vector parameters in the neural network are trained by other optimization methods, such as SGD or Adam.

For the remainder of the paper, we will not take separate layers into account and represent all the model parameters as a single matrix for simplicity of presentation. We refer to Riabinin et al. (2025) for an explanation of how to take into account every layer separately in the analysis of Muon.

## 3 LocalMuon does not Always Converge

First, we provide a lower bound showing that straightforwardly using the optimizer with the LMO as the local optimizer in FedAvg does not always converge. For simplicity, we consider the setting where all clients participate in every round and perform exactly one local update. Straightforwardly applying the optimizer with the LMO to FedAvg yields the following update rules:

$$\boldsymbol{M}_i^{(r+1)} = (1 - \alpha)\boldsymbol{M}_i^{(r)} + \alpha \nabla F_i(\boldsymbol{X}^{(r)}, \xi_i^{(r)}), \tag{1}$$

$$\boldsymbol{X}_i^{(r+1)} = \boldsymbol{X}_i^{(r)} + \eta \text{lmo}\left(\boldsymbol{M}_i^{(r+1)}\right), \tag{2}$$

$$\boldsymbol{X}^{(r+1)} = \frac{1}{n} \sum_{i=1}^{n} \boldsymbol{X}_i^{(r+1)}. \tag{3}$$

We refer to the above algorithm as LocalMuon (see Appendix C for LocalMuon with multiple local steps and partial participation). However, the above straightforward method fails to reach a stationary point due to the bias introduced by the LMO, and the optimization process stagnates. Specifically, the LMO is biased, since in general we have

$$\frac{1}{n} \sum_{i=1}^{n} \text{lmo}\left(\boldsymbol{M}_i^{(r+1)}\right) \neq \text{lmo}\left(\frac{1}{n} \sum_{i=1}^{n} \boldsymbol{M}_i^{(r+1)}\right).$$

The momentum $M_i$ is the estimation of the gradient $\nabla f_i(X)$, while the quantity of $\frac{1}{n} \sum_{i=1}^{n} \mathrm{lmo}(M_i)$ is biased and does not align with the gradient $\nabla f(X)$. This intuitively shows why LOCALMUON cannot converge to the stationary point, especially when clients have different loss functions. The following theorem formalizes this failure, with the proof deferred to Appendix D.

**Theorem 1.** *For simplicity, we consider the initialization $M_i^{(0)} = 0$. There exist convex functions $\{f_i\}_{i=1}^{n}$ such that for any $r \geq 1$ rounds, the output of LOCALMUON (Eqs. (1) to (3)) is the same as the initial parameter and does not converge to the optimal solution and satisfies the following:*

$$\|\nabla f(X^{(r)})\|^2 \geq \Omega(\zeta_\star^2),$$

*where $\zeta_\star^2 := \frac{1}{n} \sum_{i=1}^{n} \|\nabla f_i(X^\star)\|^2$ and $X^\star := \arg\min f(X)$.*

Note that LOCALMUON is a simplified version of MuLoCo (Thérien et al., 2025), where the momentum at the central server is disabled. We formally analyze only LOCALMUON, while Theorem 1 shows that the parameter stays at the initial parameters and does not converge to the stationary point. This indicates that MuLoCo also suffers from the same issue, as adding the momentum at the central server does not prevent the parameter from remaining at its initial value.

## 4 FEDMUON

In the previous section, we showed that due to the bias caused by the LMO, LOCALMUON does always converge. In this section, in Algorithm 1 we propose FEDMUON, which mitigates this issue and provably converges to the stationary point.

Instead of applying the LMO to the momentum alone, we apply the LMO to the bias corrected version of the momentum (line 8) in Algorithm 1. Similarly to SCAFFOLD (Karimireddy et al., 2020) we introduce control variates $C_i^{(r)}$ and $C^{(r)}$ to estimate the directions of the local client gradients $\nabla f_i(X^{(r)})$ and the global gradient $\nabla f(X^{(r)})$, respectively. Given that the local momentum parameters $M_i^{(r,k+1)}$ estimate local gradients $\nabla f_i(X^{(r,k)})$, the corrected update, $M_i^{(r,k+1)} - C_i^{(r)} + C^{(r)}$ is a good estimation of the full gradient $\nabla f(X^{(r,k)})$, mitigating the issue of local bias. When we remove the LMO and set $\alpha = 1$, FEDMUON recovers vanilla SCAFFOLD (Karimireddy et al., 2020). It is important to note that there are several papers that apply the momentum to SCAFFOLD (Cheng et al., 2024; Karimireddy et al., 2021), however all of them incorporate momentum at the central server, differing from our proposed FEDMUON.

---

**Algorithm 1** FEDMUON

---

1: **Input:** total number of clients $n$, number of sampled clients $S$, and the number of local steps $K$.
2: **for** $r \in \{0, 1, \cdots, R-1\}$ **do** *(at the server)*
3:     sample $S$ clients $\mathcal{S}_r \subset [n]$.
4:     send $X^{(r)}$ and $C^{(r)}$ to the sampled clients.
5:     **for** $i \in \mathcal{S}_r$ **do** *(at the clients)*
6:         $X_i^{(r,0)} \leftarrow X^{(r)}$ and $M_i^{(r,0)} \leftarrow M_i^{(r-1,K)}$
7:         **for** $k = 0, 1, \cdots, K-1$ **do**
8:             $M_i^{(r,k+1)} \leftarrow (1-\alpha) M_i^{(r,k)} + \alpha \nabla F_i(X_i^{(r,k)}; \xi_i^{(r,k)})$.
9:             $X_i^{(r,k+1)} \leftarrow X_i^{(r,k)} + \eta \mathrm{lmo}\left(M_i^{(r,k+1)} - C_i^{(r)} + C^{(r)}\right)$.
10:         **end for**
11:         $C_i^{(r+1)} \leftarrow M_i^{(r,K)}$
12:         send $X_i^{(r,K)}$ and $C_i^{(r+1)}$ to the central server.
13:     **end for** *(end clients, back to the server)*
14:     **for** $i \in [n] \setminus \mathcal{S}_r$ **do**
15:         $C_i^{(r+1)} \leftarrow C_i^{(r)}$ and $M_i^{(r,K)} \leftarrow M_i^{(r-1,K)}$.
16:     **end for**
17:     $C^{(r+1)} \leftarrow C^{(r)} + \frac{1}{N} \sum_{i \in \mathcal{S}_r} \left(C_i^{(r+1)} - C_i^{(r)}\right)$.
18:     $X^{(r+1)} \leftarrow \frac{n-S}{n} X^{(r)} + \frac{1}{n} \sum_{i \in \mathcal{S}_r} X_i^{(r,K)}$.
19: **end for**

---

## 5 CONVERGENCE ANALYSIS

### 5.1 ASSUMPTIONS

We first summarize the assumptions that we use in our theoretical results. As is common in the prior literature analyzing optimizers with LMO (Pethick et al., 2025; Riabinin et al., 2025), we use the following smoothness assumption. Note that the norm here is the same as the one used in the LMO.

**Assumption 1.** *There exists $L \geq 0$ so that it holds for any $\boldsymbol{X}, \boldsymbol{Y} \in \mathcal{X}$*

$$\|\nabla f_i(\boldsymbol{X}) - \nabla f_i(\boldsymbol{Y})\|_\star \leq L\|\boldsymbol{X} - \boldsymbol{Y}\|.$$

Since we consider non-Euclidean norms, we measure gradient differences in the dual norm, while parameter differences are measured in the primal norm (cf. Nesterov, 2018; Xie & Li, 2024). However, as shown in Remark 1, any two norms are equivalent in finite dimensions, and thus the class of functions satisfying Assumption 1 and the conventional smoothness assumptions (formulated for Euclidean norms) is the same (see Remark 2).

**Remark 1** ((Conway, 2019, Theorem 3.1)). *If $\mathcal{X}$ is a finite-dimensional vector space over $\mathbb{F}$, then for any two norms $\|\cdot\|_p$ and $\|\cdot\|_q$, there exist $c, C \geq 0$ such that $c\|\boldsymbol{X}\|_p \leq \|\boldsymbol{X}\|_q \leq C\|\boldsymbol{X}\|_p$ for all $\boldsymbol{X} \in \mathcal{X}$.*

**Remark 2.** *If it holds that $\|\nabla f_i(\boldsymbol{X}) - \nabla f_i(\boldsymbol{Y})\| \leq CL\|\boldsymbol{X} - \boldsymbol{Y}\|$ for any $\boldsymbol{X}, \boldsymbol{Y} \in \mathcal{X}$, then $f_i$ satisfies Assumption 1 where $C := \sup_{\boldsymbol{X} \in \mathcal{X}} \frac{\|\boldsymbol{X}\|_\star}{\|\boldsymbol{X}\|}$. If $f_i$ satisfies Assumption 1, it holds that $\|\nabla f_i(\boldsymbol{X}) - \nabla f_i(\boldsymbol{Y})\| \leq \frac{L}{c}\|\boldsymbol{X} - \boldsymbol{Y}\|$ for any $\boldsymbol{X}, \boldsymbol{Y} \in \mathcal{X}$ where $c := \sup_{\boldsymbol{X} \in \mathcal{X}} \frac{\|\boldsymbol{X}\|}{\|\boldsymbol{X}\|_\star}$.*

For the analysis of FEDMUON, we often use the trace norm and Frobenius norm. The following inequality holds between the Frobenius norm and the trace norm.

**Example 1.** *For any $\boldsymbol{X} \in \mathbb{R}^{d_1 \times d_2}$, it holds that $\|\boldsymbol{X}\|_F \leq \|\boldsymbol{X}\|_{trace} \leq \sqrt{\min\{d_1, d_2\}}\|\boldsymbol{X}\|_F$.*

For the stochastic gradient noise, we use the following assumption, which is quite common in the optimization literature, e.g., (Bubeck, 2015).

**Assumption 2.** *The stochastic gradient is unbiased, i.e., $\mathbb{E}[\nabla F_i(\boldsymbol{X}; \xi_i)] = \nabla f_i(\boldsymbol{X})$ for any $\boldsymbol{X} \in \mathcal{X}$. Then, there exists $\sigma \geq 0$ so that it holds for any $\boldsymbol{X} \in \mathcal{X}$*

$$\mathbb{E}_{\xi_i \sim \mathcal{D}_i} \|\nabla F_i(\boldsymbol{X}; \xi_i) - \nabla f_i(\boldsymbol{X})\|_F^2 \leq \sigma^2.$$

### 5.2 CONVERGENCE RESULT

We provide the convergence rate of FEDMUON in Theorem 2. For simplicity, we present the results for the special case $S = n$, where all clients participate during the training. The general case with arbitrary $S$ is provided in Lemma 11 in Appendix E. The proof is deferred to Appendix E.

**Theorem 2.** *Consider Algorithm 1. We define $\boldsymbol{X}^{(r,k)} = \frac{1}{n}\sum_{i=1}^n \boldsymbol{X}_i^{(r,k)}$. Note that $\boldsymbol{X}^{(r+1)} = \boldsymbol{X}^{(r,K)}$. Suppose that $n = S$ and Assumptions 1 and 2 hold, $\boldsymbol{C}_i^{(0)} := \boldsymbol{M}_i^{(0,0)}$ and $\boldsymbol{C}^{(0)} := \frac{1}{n}\sum_{i=1}^n \boldsymbol{C}_i^{(0)}$, there exists $\eta$ and $\alpha$ so that it satisfies*

$$\frac{1}{RK}\sum_{r=0}^{R-1}\sum_{k=0}^{K-1}\mathbb{E}\left\|\nabla f(\boldsymbol{X}^{(r,k)})\right\|_\star \leq \mathcal{O}\left(\left(\frac{Lr_0\tilde{\sigma}^2}{nRK}\right)^{\frac{1}{4}} + \left(\frac{Lr_0\tilde{\sigma}}{R\sqrt{K}}\right)^{\frac{1}{3}} + \left(\frac{Lr_0}{R}\right)^{\frac{1}{2}}\right.$$
$$\left. + \tilde{\sigma}_0\left[\frac{1}{R} + \left(\frac{\tilde{\sigma}^2 K}{Lr_0 Rn}\right)^{\frac{1}{2}} + \left(\frac{\tilde{\sigma}^2 K^2}{Lr_0 R^2}\right)^{\frac{1}{3}}\right]\right),$$

*where $r_0 := f(\boldsymbol{X}^{(0)}) - f^\star$, $\rho := \sup_{\boldsymbol{X} \in \mathcal{X}} \frac{\|\boldsymbol{X}\|_\star}{\|\boldsymbol{X}\|_F}$, $\tilde{\sigma} := \rho\sigma$, $\tilde{\sigma}_0 := \rho\sigma_0$, and $\sigma_0^2 := \frac{1}{n}\sum_{i=1}^n \mathbb{E}\|\boldsymbol{M}_i^{(0,0)} - \nabla f_i(\boldsymbol{X}^{(0)})\|_F^2$.*

**Discussion:** Unlike LOCALMUON, Theorem 2 shows that FEDMUON can mitigate the issue that the LMO is a biased operator and can converge to the stationary point. The dominant term is $\mathcal{O}(\frac{Lr_0\tilde{\sigma}^2}{nRK})^{\frac{1}{4}}$,

---

**Algorithm 2** Newton-Schulz iteration

---

1: **Input:** matrix $\boldsymbol{G}$ and hyperparameters $a, b, c \in \mathbb{R}$.
2: $\boldsymbol{G}^{(0)} \leftarrow \frac{\boldsymbol{G}}{\|\boldsymbol{G}\|_F}$.
3: **for** $t \in \{0, 1, \cdots, T-1\}$ **do**
4: $\quad \boldsymbol{G}^{(t+1)} \leftarrow a\boldsymbol{G}^{(t)} + b(\boldsymbol{G}^{(t)}\boldsymbol{G}^{(t)^\top})\boldsymbol{G}^{(t)} + c(\boldsymbol{G}^{(t)}\boldsymbol{G}^{(t)^\top})^2\boldsymbol{G}^{(t)}$.
5: **end for**
6: **Retern** $-\boldsymbol{G}^{(T)}$

---

which is almost the same as the terms appearing in the rate of FedAvg and SCAFFOLD (Karimireddy et al., 2020), and the convergence rate is improved as the number of clients $n$ increases. The only difference is that the convergence rate of FEDMUON depends on $\rho$, while this is because Theorem 2 analyzes the dual norm of the gradient. For instance, when the norm is the Frobenius norm, the dual norm is also the Frobenius norm and $\rho = 1$. The last three terms arise from the initial error $\sigma_0$, which diminish faster than the other terms as the number of rounds $R$ increases.

We consider the case where the parameter space is $\mathcal{X} = \mathbb{R}^{d_1 \times d_2}$ and the spectral norm is used, as in Muon (Liu et al., 2025a). Since the dual of the spectral norm is the trace norm, we have $\|\nabla f(\boldsymbol{X})\|_F \leq \|\nabla f(\boldsymbol{X})\|_{\text{trace}}$. Consequently, FEDMUON can converge faster than SCAFFOLD in certain cases. For instance, if the stochastic noise $\sigma$ is sufficiently small, FEDMUON converges as:

$$\frac{1}{RK}\sum_{r=0}^{R-1}\sum_{k=0}^{K-1}\mathbb{E}\left\|\nabla f(\boldsymbol{X}^{(r,k)})\right\|_{\text{trace}} \leq \mathcal{O}\left(\left(\frac{r_0 L}{R}\right)^{\frac{1}{2}}\right),$$

and SCAFFOLD converges as follows (see Theorem 3 in (Karimireddy et al., 2020)):

$$\frac{1}{RK}\sum_{r=0}^{R-1}\sum_{k=0}^{K-1}\mathbb{E}\left\|\nabla f(\boldsymbol{X}^{(r,k)})\right\|_F \leq \mathcal{O}\left(\left(\frac{r_0 L_F}{R}\right)^{\frac{1}{2}}\right).$$

where $L_F$ refers to the smoothness of $f_i$ with respect to the Frobenius norm. Thus, when $L = \sup_{i \in [n], \boldsymbol{X}, \|\boldsymbol{U}\|_{\text{sp}} \leq 1}\langle \boldsymbol{U}, \nabla^2 f_i(\boldsymbol{X})\boldsymbol{U}\rangle \approx L_F$, i.e., when the Hessians have a few dominant singular values—equivalently, when they are approximately low-rank, then FEDMUON can converge faster than SCAFFOLD. More precisely, the terms on the right-hand side are the same, and the only difference is the choice of the norm. We stress that Theorem 2 does not claim FEDMUON always converges faster, but it does suggest that in certain cases FEDMUON can outperform. This helps explain the strong empirical performance of Muon and FEDMUON.

## 6 FEDMUON WITH INEXACT LMO

In the previous section, we considered the general case with an arbitrary norm and exact LMO. Here, we focus on the spectral norm, as in Muon (Liu et al., 2025a), and analyze FEDMUON when the LMO is only approximately solved via the Newton–Schulz iteration. Then, thanks to the special property of spectral norm and Newton-Schulz iteration, we reveal that FEDMUON can converge to the stationary point regardless of how accurately we solve the LMO.

With the spectral norm, the LMO takes the following form:

$$\text{lmo}_{\text{muon}}(\boldsymbol{X}) \coloneqq \underset{\boldsymbol{Y} \in \{\boldsymbol{Y} \in \mathbb{R}^{d_1 \times d_2} \,|\, \|\boldsymbol{Y}\|_{\text{sp}} \leq 1\}}{\arg\min} \langle \boldsymbol{X}, \boldsymbol{Y} \rangle,$$

Let the singular value decomposition of $\boldsymbol{X}$ be $\boldsymbol{U}\boldsymbol{\Sigma}\boldsymbol{V}$. Then the LMO output is $-\boldsymbol{U}\boldsymbol{V}$, but computing this exactly is computationally expensive. To address this, Liu et al. (2025a) proposed approximating the LMO via a fixed number of Newton–Schulz iterations (Schulz, 1933)(e.g., 5). The update rule of the Newton-Schulz iteration is described in Algorithm 2. Since the procedure involves only matrix multiplications, it can be efficiently executed on a GPU. In the following, we analyze the convergence of FEDMUON when the LMO is solved approximately using Newton–Schulz iterations and characterize how inexactness impacts convergence.

Under the same assumption as in Theorem 2, we provide the convergence rate when we run Newton-Schulz iteration $T$ times to solve the LMO approximately and show how $T$ affects convergence. For

simplicity, we set $n = S$ and we use $a = \frac{15}{8}, b = -\frac{5}{4}$ and $c = \frac{3}{8}$ for the Newton-Schulz iteration, following the hyperparameter setting mentioned in Amsel et al. (2025). For the general case of arbitrary $S$ we refer to Lemma 14 in Appendix F.

**Theorem 3.** *Consider Algorithm 1 with the spectral norm and suppose that the LMO in line 8 is solved approximately using Algorithm 2 with $a = \frac{15}{8}, b = -\frac{5}{4}$, and $c = \frac{3}{8}$. We define $\boldsymbol{X}^{(r,k)} = \frac{1}{n} \sum_{i=1}^{n} \boldsymbol{X}_i^{(r,k)}$. Note that $\boldsymbol{X}^{(r+1)} = \boldsymbol{X}^{(r,K)}$. Suppose that $n = S$ and Assumptions 1 and 2 hold, $\boldsymbol{C}_i^{(0)} := \boldsymbol{M}_i^{(0,0)}$ and $\boldsymbol{C}^{(0)} := \frac{1}{n} \sum_{i=1}^{n} \boldsymbol{C}_i^{(0)}$. Then, for any number of Newton-Schulz iteration $T \geq 0$, there exists $\eta$ and $\alpha$ so that it satisfies*

$$\frac{1}{RK} \sum_{r=0}^{R-1} \sum_{k=0}^{K-1} \mathbb{E} \left\| \nabla f(\boldsymbol{X}^{(r,k)}) \right\|_p \leq \mathcal{O} \left( \left( \frac{Lr_0 \tilde{\sigma}^2}{nRK} \right)^{\frac{1}{4}} + \left( \frac{Lr_0 \tilde{\sigma}}{R\sqrt{K}} \right)^{\frac{1}{3}} + \left( \frac{Lr_0}{R} \right)^{\frac{1}{2}} \right.$$

$$\left. + \tilde{\sigma}_0 \left[ \frac{1}{R} + \left( \frac{\tilde{\sigma}^2 K}{Lr_0 Rn} \right)^{\frac{1}{2}} + \left( \frac{\tilde{\sigma}^2 K^2}{Lr_0 R^2} \right)^{\frac{1}{3}} \right] \right),$$

*where $r_0 := f(\boldsymbol{X}^{(0)}) - f^{\star}, \rho := \sqrt{\min\{d_1, d_2\}}, \tilde{\sigma} := \rho\sigma, \tilde{\sigma}_0 := \rho\sigma_0$, and $\sigma_0^2 := \frac{1}{n} \sum_{i=1}^{n} \mathbb{E} \|\boldsymbol{M}_i^{(0,0)} - \nabla f_i(\boldsymbol{X}^{(0)})\|_F^2$. Then, $p$ is defined as follows:*

$$p := 1 + \frac{\log \left( 1 - (1 - \kappa)^{1.5^T} \right)}{\log \kappa},$$

$$\kappa := \min_{j,i,r,k} \frac{s_{j,i,r,k}}{\sqrt{\sum_{j'} s_{j',i,r,k}^2}} \quad (> 0),$$

*where $\{s_{j,i,r,k}\}_j$ are non-zero singular values of $\boldsymbol{M}_i^{(r,k+1)} - \boldsymbol{C}_i^{(r)} + \boldsymbol{C}^{(r)}$.*

**Remark 3.** *When $T = 0$, $p = 2$. As $T$ increase, $p$ monotonically decreases to 1 for any $\kappa > 0$.*

**Remark 4.** *Recall that $\| \cdot \|_p$ is the Schatten $p$-norm. For any $1 \leq p \leq q$, we have $\|\boldsymbol{A}\|_q \leq \|\boldsymbol{A}\|_p$. Then, $\|\boldsymbol{A}\|_p$ becomes $\|\boldsymbol{A}\|_{trace}$ and $\|\boldsymbol{A}\|_F$ when $p = 1$ and $p = 2$, respectively.*

**Discussion:** Surprisingly, the above theorem shows that FEDMUON converges to the stationary point, regardless of how many times we run the Newton-Schulz iteration. The only difference between the case when we solve the LMO exactly, i.e., Theorem 2, and the case when we solve the LMO approximately, i.e., Theorem 3, is that Theorem 2 establishes the convergence in the trace norm of the gradient $\|\nabla f(\boldsymbol{X})\|_{\text{trace}} (= \|\nabla f(\boldsymbol{X})\|_1)$,[1] while Theorem 3 establishes the convergence in the Schatten $p$-norm $\|\nabla f(\boldsymbol{X})\|_p$. We recover the convergence rate of Theorem 2 by setting $T \to \infty$, and therefore have $p \to 1$. Since we have $\|\boldsymbol{A}\|_q \leq \|\boldsymbol{A}\|_p$ when $1 \leq p \leq q$, Theorem 3 implies that FEDMUON can converge faster when we increase the number of Newton-Schulz iterations $T$. More specifically, since it holds that $\|\boldsymbol{A}\|_1 \leq \sqrt{\min\{d_1, d_2\}}\|\boldsymbol{A}\|_2$, solving the LMO accurately can improve the convergence rate by up to a factor of $\sqrt{\min\{d_1, d_2\}}$. In our experiments, we will demonstrate that FEDMUON can train neural networks even if $T = 0$, while FEDMUON can achieve higher accuracy as $T$ increases (see Section 7.2). These observations are consistent with Theorem 3.

The quantity of $(1 - \kappa)^{1.5^T}$ in the definition of $p$ measures how fast the Newton-Schulz iteration converges. If we consider the worst case, $\kappa$ could be arbitrarily close to zero, and thus a large $T$ would be required to sufficiently decrease $p$. However, the main implication of Theorem 3 is that increasing $T$ leads to an improved convergence rate. Indeed, our experiments show that even increasing $T$ from 0 to 1 dramatically improves accuracy (see Section 7.2).

**Comparison with Existing Analysis with Inexact LMO:** There are many papers that analyzed the convergence rate of Muon, while most of them assumed that the LMO is exactly solved (Pethick et al., 2025; Riabinin et al., 2025; Shen et al., 2025). The only study analyzing the rate with an inexact LMO is Refael et al. (2025). However, they also assumed that we run Newton-Schulz iterations a certain number of times (see Lemma 3.3 and Remark 3.6 in (Refael et al., 2025)). Compared with these prior analyses, our novel analysis provides a stronger claim that FEDMUON can converge to the stationary point for any number of the Newton-Schulz iterations $T \geq 0$. Furthermore, it is first observed by Theorem 3 that the different norms of the gradient are bounded depending on $T$.

---

[1]When $\| \cdot \|$ is the spectral norm, its dual norm is the trace norm.

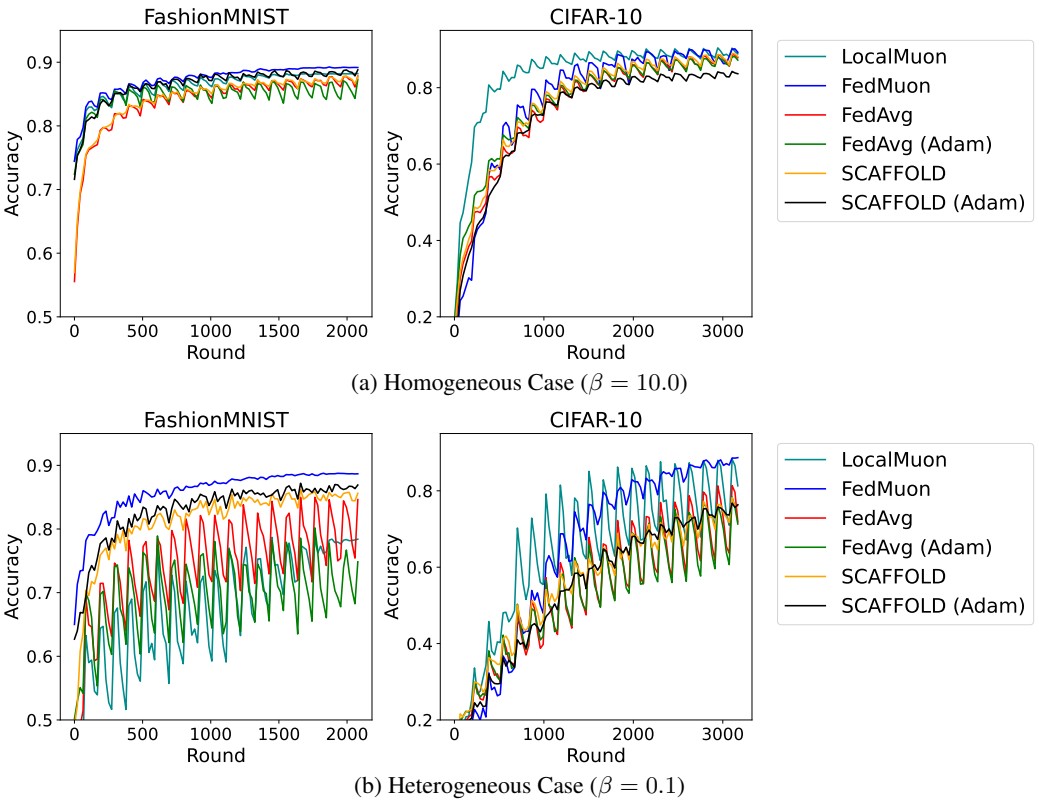

(a) Homogeneous Case ($\beta = 10.0$)

(b) Heterogeneous Case ($\beta = 0.1$)

Figure 1: Training curves of various methods. For all settings, FEDMUON can achieve higher test accuracy than other methods.

**Proof Sketch:** In the following, we provide an intuition for why FEDMUON can converge for any $T \geq 0$. If we solve the LMO exactly, we have

$$\langle \boldsymbol{G}, \text{lmo}_{\text{muon}}(\boldsymbol{G}) \rangle = -\|\boldsymbol{G}\|_{\text{trace}}, \qquad \|\text{lmo}_{\text{muon}}(\boldsymbol{G})\|_{\text{sp}} \leq 1. \qquad (4)$$

The first equality holds from the definition of the dual norm (see Lemma 1), and the second inequality holds since the solution of the LMO satisfies the constraint. Then, the output of the Newton-Schulz iteration satisfies the following (see Lemma 12):

$$-\|\boldsymbol{G}\|_{\text{trace}} \leq \langle \boldsymbol{G}, -\boldsymbol{G}^{(T)} \rangle \leq -\|\boldsymbol{G}\|_p, \qquad \left\|-\boldsymbol{G}^{(T)}\right\|_{\text{sp}} \leq 1. \qquad (5)$$

The above inequality indicates that even if we run the Newton-Schulz iteration only a few times to solve the LMO approximately, the output of the Newton-Schulz iteration is a proper direction to minimize the loss function, and FEDMUON can converge to the stationary point. For instance, when $T = 0$, the output of Newton-Schulz iteration is $-\frac{\boldsymbol{G}}{\|\boldsymbol{G}\|_F}$, which corresponds to the normalized gradient, and it is natural that FEDMUON can converge to the stationary point when $T = 0$. Then, if we run the Newton-Schulz iteration $T$ times, the output of the Newton-Schulz iteration comes close to the exact solution of LMO and remains a proper direction to minimize the loss function. Thanks to this property, FEDMUON can converge to the stationary point for any number of Newton-Schulz iterations $T$.

## 7 EXPERIMENT

### 7.1 FEDERATED LEARNING TASKS

**Setup:** We used FashionMNIST (Xiao et al., 2017) and CIFAR-10 (Krizhevsky, 2009) as training datasets, and used LeNet (Lecun et al., 1998) for Fashion MNIST and ResNet-18 (He et al., 2016) for CIFAR-10. Following the prior paper (Hsieh et al., 2020), we used Group Normalization (Wu & He,

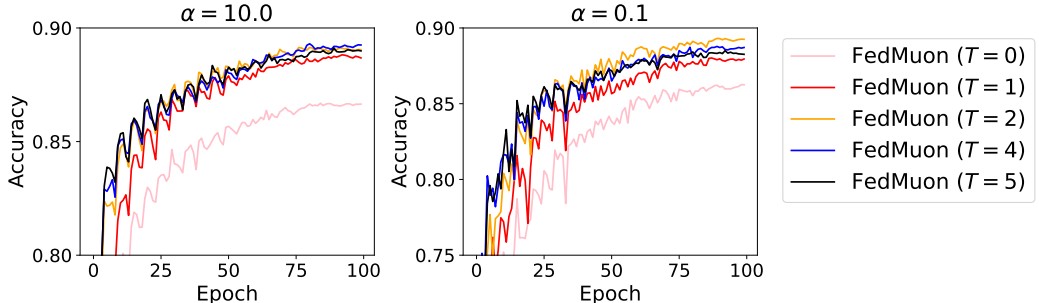

Figure 2: Training curves of FEDMUON with various number of Newton-Schulz iterations. We used FashionM-NIST and LeNet.

2018) instead of Batch Normalization (Ioffe & Szegedy, 2015) for ResNet-18. We set the number of clients $n$ to 16 and sampled $S = 8$ clients every round. We set the number of local steps $K$ to 5 and set the number of epochs to 100 and 200 for FashionMNIST and CIFAR-10, respectively. Following the prior paper (Hsu et al., 2019), we distributed the training dataset to clients by using Dirichlet distributions with hyperparameter $\beta$. As $\beta$ approaches zero, each clients come to have a different training dataset. We tuned the stepsize by grid search. See Appendix G for details. The experiments were repeated with two different seed values, and we reported the average.

**Comparison Methods:** We compared the following methods: (1) FedAvg (McMahan et al., 2017): We used Momentum SGD as the optimizer. (2) FedAvg (Adam): We used Adam as the optimizer of FedAvg. (3) SCAFFOLD (Karimireddy et al., 2020): We used Momentum SGD as the optimizer. (4) SCAFFOLD (Adam): We used Adam as the optimizer of SCAFFOLD. (5) FEDMUON: Our proposed method. Following the suggestion of Liu et al. (2025a), we changed the scale of the stepsize per layer, depending on the dimension.

**Results:** We show the results in Fig. 1. The results indicate that FEDMUON can perform the best for all settings. By comparing FEDMUON with FedAvg (Adam) and SCAFFOLD (Adam), FEDMUON achieved the highest accuracy, which can demonstrate that Muon is also beneficial in the federated learning setting. By comparing FEDMUON and LOCALMUON, LOCALMUON performed well in the homogeneous setting, but did not match the performance of FEDMUON in the heterogeneous setting. This observation is consistent with the discussion in Section 3, where we show that LOCALMUON does not converge to the stationary point in the heterogeneous setting. These observations were consistent with Theorem 1.

## 7.2 EFFECT OF INEXACT LMO

Next, we evaluate how the number of Newton-Schulz iterations $T$ affects the performance. Figure 2 shows the training curves of FEDMUON with different $T$. In the homogeneous setting, the highest accuracy was achieved when $T = 4$, and in the heterogeneous setting, the highest accuracy was achieved when $T = 2$. Thus, we can observe that solving the LMO accurately can improve the performance. Notably, FEDMUON already worked with $T = 0$, but increasing $T$ from 0 to 1 led to a significant improvement in accuracy. These observations were consistent with Theorem 3, which shows that FEDMUON can converge for any $T$ and converge faster as $T$ increases.

## 8 CONCLUSION

In this paper, we study the federated learning methods with the LMO and propose FEDMUON. We first propose directly plugging the optimization methods with the LMO into FedAvg, which we referred to as LOCALMUON, and show that LOCALMUON cannot converge to the stationary point since the LMO is a biased operator. We then propose FEDMUON to solve this issue and show that FEDMUON can converge to the stationary point. We analyze the convergence rate of FEDMUON and reveal how the approximate solution of the LMO affects the convergence behavior. Notably, we show that FEDMUON can converge for any number of Newton-Schulz iterations, and FEDMUON can converge faster as we solve the LMO more accurately. Throughout the experiments, we demonstrated the effectiveness of FEDMUON and verified our theoretical discovery.

ACKNOWLEDGEMENTS

This paper was supported by JSPS KAKENHI Grant Number 23KJ1336.

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

# A  LLM USAGE

We used LLM for proofreading, and it did not contribute to the content of the paper itself.

# B  RELATED WORK

**Federated Learning:**   The simplest algorithm for federated learning is FedAvg (McMahan et al., 2017; Stich, 2019). The main challenge of federated learning is reducing communication between the central server and clients. Various techniques such as client sampling (Gu et al., 2021; Chen et al., 2022; Zhang et al., 2023), multiple local steps (Woodworth et al., 2020; Koloskova et al., 2020; Jiang et al., 2024a;b), and communication compression (Alistarh et al., 2017; Stich et al., 2018; Karimireddy et al., 2019; Vogels et al., 2019; He et al., 2023; Gao et al., 2024) have been studied to reduce the communication costs. However, FedAvg still requires a huge amount of communication when clients have different training datasets. Many papers proposed federated learning methods that are robust to data heterogeneity (Karimireddy et al., 2020; Jiang et al., 2024a;b). The seminal work is SCAFFOLD (Karimireddy et al., 2020), which can converge regardless of data heterogeneity. Besides these methods, asynchronous methods (Koloskova et al., 2022; Mishchenko et al., 2022; Islamov et al., 2024) and decentralized methods (Nedić et al., 2017; Tang et al., 2018b;a; Koloskova et al., 2020; Takezawa et al., 2023) have been widely studied to further improve the efficiency.

**Adaptive Optimization Methods:**   Using adaptive optimization methods is standard for training neural networks efficiently (Amari, 1998; Ward et al., 2020; Duchi et al., 2011; Kingma & Ba, 2017; Loshchilov & Hutter, 2019; Zaheer et al., 2018; Zhuang et al., 2020; Defazio et al., 2024; Rodomanov et al., 2024). Over the last decade, Adam (Kingma & Ba, 2017) and AdamW (Loshchilov & Hutter, 2019) are the most widely used, but recently, Shampoo (Gupta et al., 2018) won the External Tuning Task of AlgoPerf (Dahl et al., 2025) and is attracting considerable attention (Shi et al., 2023; Vyas et al., 2025; Ishikawa & Karakida, 2024). Muon (Liu et al., 2025a) can be regarded as the simplified version of Shampoo, and many papers have demonstrated that Muon can train neural networks faster than Adam, AdamW, and Shampoo (Liu et al., 2025a; Pethick et al., 2025; Amsel et al., 2025; Liu et al., 2025b; Ma et al., 2025; Amsel et al., 2025; Grishina et al., 2025). Using Muon in distributed environments is one of the popular topics (Thérien et al., 2025; Ahn et al., 2025). Specifically, Ahn et al. (2025) proposed a method to solve the LMO in a distributed way, and Thérien et al. (2025) proposed MuLoCo, which extends Muon by allowing clients to perform several steps before averaging the parameters as in LOCALMUON. However, since they consider settings where all clients have the same dataset, their objective differs from ours. As we explained in Section 3, because the LMO is a biased operator, bias correction mechanisms used in FEDMUON are necessary in federated learning, in which clients have different training datasets.

## C  PSEUDO CODE

---

**Algorithm 3** FedAvg (McMahan et al., 2017)

---

1: **Input:** the total number of clients $n$, the number of sampled clients $S$, and the number of local steps $K$.
2: **for** $t \in \{0, 1, \cdots, T\}$ **do** *(at the server)*
3:     sample $S$ clients $\mathcal{S}_r \subset [n]$.
4:     **for** $i \in \mathcal{S}_r$ **do** *(at the clients)*
5:         $\boldsymbol{X}_i^{(r,0)} \leftarrow \boldsymbol{X}^{(r)}$.
6:         **for** $k = 0, 1, \cdots, K-1$ **do**
7:             $\boldsymbol{X}_i^{(r,k+1)} \leftarrow \boldsymbol{X}_i^{(r,k)} - \eta \nabla F_i(\boldsymbol{X}_i^{(r,k)}; \xi_i^{(r,k)})$.
8:         **end for**
9:     **end for** *(end clients, back to the server)*
10:    $\boldsymbol{X}^{(r+1)} \leftarrow \frac{n-S}{n}\boldsymbol{X}^{(r)} + \frac{1}{n}\sum_{i \in \mathcal{S}_r} \boldsymbol{X}_i^{(r,K)}$.
11: **end for**

---

**Algorithm 4** LocalMuon

---

1: **Input:** the total number of clients $n$, the number of sampled clients $S$, and the number of local steps $K$.
2: **for** $r \in \{0, 1, \cdots, R-1\}$ **do** *(at the server)*
3:     sample $S$ clients $\mathcal{S}_r \subset [n]$.
4:     **for** $i \in \mathcal{S}_r$ **do** *(at the clients)*
5:         $\boldsymbol{X}_i^{(r,0)} \leftarrow \boldsymbol{X}^{(r)}$ and $\boldsymbol{M}_i^{(r,0)} \leftarrow \boldsymbol{M}_i^{(r-1,K)}$
6:         **for** $k = 0, 1, \cdots, K-1$ **do**
7:             $\boldsymbol{M}_i^{(r,k+1)} \leftarrow (1-\alpha)\boldsymbol{M}_i^{(r,k)} + \alpha \nabla F_i(\boldsymbol{X}_i^{(r,k)}; \xi_i^{(r,k)})$.
8:             $\boldsymbol{X}_i^{(r,k+1)} \leftarrow \boldsymbol{X}_i^{(r,k)} + \eta\mathrm{lmo}\left(\boldsymbol{M}_i^{(r,k+1)}\right)$.
9:         **end for**
10:       $\boldsymbol{C}_i^{(r+1)} \leftarrow \boldsymbol{M}_i^{(r,K)}$
11:     **end for**
12:     **for** $i \in [n] \setminus \mathcal{S}_r$ **do**
13:         $\boldsymbol{M}_i^{(r,K)} \leftarrow \boldsymbol{M}_i^{(r-1,K)}$.
14:     **end for** *(end clients, back to the server)*
15:    $\boldsymbol{X}^{(r+1)} \leftarrow \frac{n-S}{n}\boldsymbol{X}^{(r)} + \frac{1}{n}\sum_{i \in \mathcal{S}_r} \boldsymbol{X}_i^{(r,K)}$.
16: **end for**

---

# D  PROOF OF THEOREM 1

*Proof.* We consider the setting where $n = 2$, $d = 1$, and the norm is the Euclidean norm. In this case, we have

$$\text{lmo}(x) = \frac{x}{|x|}.$$

Then, we consider the case where $f_1$ and $f_2$ are defined as follows:

$$f_1(x) := \frac{x^2}{2},$$
$$f_2(x) := \frac{(x+a)^2}{2}.$$

When $\boldsymbol{M}_i^{(0)} = 0$ and $\boldsymbol{X}^{(0)} = -\frac{a}{4}$, we have

$$\boldsymbol{M}_1^{(1)} = -\frac{\alpha a}{4},$$
$$\boldsymbol{M}_2^{(1)} = \frac{3\alpha a}{4},$$
$$\text{lmo}(\boldsymbol{M}_1^{(1)}) + \text{lmo}(\boldsymbol{M}_2^{(1)}) = 0,$$

where we use $\alpha \in (0, 1]$.

Thus, the parameter does not change, i.e., $\boldsymbol{X}^{(1)} = \boldsymbol{X}^{(0)}$. For the next round, we have

$$\boldsymbol{M}_1^{(2)} = -\frac{a}{4}\left(\alpha + \alpha(1-\alpha)\right),$$
$$\boldsymbol{M}_2^{(2)} = \frac{3a}{4}\left(\alpha + \alpha(1-\alpha)\right).$$

Then, since it holds the following as in the first round:

$$\text{lmo}(\boldsymbol{M}_1^{(2)}) + \text{lmo}(\boldsymbol{M}_2^{(2)}) = 0.$$

The parameter does not change. Due to the above discussion, the parameter does not change for any $r$. Now, we have

$$\|\nabla f(\boldsymbol{X}^{(r)})\|^2 = \frac{a^2}{16}. \tag{6}$$

Then, using $\frac{1}{2}\sum_{i=1}^2 \|\nabla f(\boldsymbol{X}^\star)\|^2 = \frac{5a^2}{16}$, we obtain the desired result. $\qquad\square$

# E  PROOF OF THEOREM 2

## E.1  NOTATION

In this section, we use the following notation.

$$\boldsymbol{X}^{(r,k)} = \frac{n-S}{n}\boldsymbol{X}^{(r)} + \frac{1}{n}\sum_{i\in\mathcal{S}_r}\boldsymbol{X}_i^{(r,k)}, \tag{7}$$

$$\boldsymbol{G}_i^{(r,k+1)} = \boldsymbol{M}_i^{(r,k+1)} - \boldsymbol{C}_i^{(r)} + \boldsymbol{C}^{(r)}, \tag{8}$$

$$\boldsymbol{D}_i^{(r,k+1)} = \mathrm{lmo}\left(\boldsymbol{G}_i^{(r,k+1)}\right). \tag{9}$$

## E.2  USEFUL LEMMA

**Lemma 1.** *For any $\boldsymbol{X} \in \mathcal{X}$, we have*

$$\langle \boldsymbol{X}, lmo(\boldsymbol{X})\rangle = -\|\boldsymbol{X}\|_\star.$$

*Proof.* From the definition of $\mathrm{lmo}(\cdot)$, we have

$$
\begin{aligned}
\langle \boldsymbol{X}, \mathrm{lmo}(\boldsymbol{X})\rangle &= \min_{\boldsymbol{Y}\in\{\boldsymbol{Y}\in\mathcal{X}|\|\boldsymbol{Y}\|\leq 1\}}\langle \boldsymbol{X}, \boldsymbol{Y}\rangle \\
&= -\max_{\boldsymbol{Y}\in\{\boldsymbol{Y}\in\mathcal{X}|\|\boldsymbol{Y}\|\leq 1\}}\langle -\boldsymbol{X}, \boldsymbol{Y}\rangle \\
&= -\|-\boldsymbol{X}\|_\star \\
&= -\|\boldsymbol{X}\|_\star.
\end{aligned}
$$

$\square$

**Lemma 2.** *For any $k \geq 0$, $R \geq 0$ and $\alpha \in (0, 1]$, we have*

$$\sum_{r=0}^{R} k(1-\alpha)^{kr} \leq \frac{1}{\alpha} + k.$$

*Proof.* We have

$$\sum_{r=0}^{R} k(1-\alpha)^{kr} \leq \frac{k}{1-(1-\alpha)^k}.$$

Then, using $(1-\alpha)^k \leq e^{-\alpha k} \leq \frac{1}{1+\alpha k}$, we obtain the desired result. $\square$

**Lemma 3.** *For any $\boldsymbol{A}, \boldsymbol{B} \in \mathcal{X}$, we have*

$$\langle \boldsymbol{A}, \boldsymbol{B}\rangle \leq \|\boldsymbol{A}\|\|\boldsymbol{B}\|_\star.$$

*Proof.* We have

$$\langle \boldsymbol{A}, \boldsymbol{B}\rangle = \|\boldsymbol{A}\|\left\langle \frac{\boldsymbol{A}}{\|\boldsymbol{A}\|}, \boldsymbol{B}\right\rangle \leq \|\boldsymbol{A}\|\|\boldsymbol{B}\|_\star.$$

$\square$

**Lemma 4.** *Suppose that Assumption 1 holds. Then, it holds that for any $\boldsymbol{X}, \boldsymbol{Y} \in \mathcal{X}$,*

$$f_i(\boldsymbol{X}) \leq f_i(\boldsymbol{Y}) + \langle \nabla f_i(\boldsymbol{Y}), \boldsymbol{X} - \boldsymbol{Y}\rangle + \frac{L}{2}\|\boldsymbol{X} - \boldsymbol{Y}\|. \tag{10}$$

*Proof.* Using the Fundamental Theorem of Calculus, we have

$$
\begin{aligned}
f(\boldsymbol{X}) &= f(\boldsymbol{Y}) + \int_{t=0}^{1} \langle \nabla f(\boldsymbol{Y} + t(\boldsymbol{X} - \boldsymbol{Y})), \boldsymbol{X} - \boldsymbol{Y} \rangle \, dt \\
&= f(\boldsymbol{Y}) + \langle \nabla f(\boldsymbol{Y}), \boldsymbol{Y} - \boldsymbol{X} \rangle + \int_{t=0}^{1} \langle \nabla f(\boldsymbol{Y} + t(\boldsymbol{X} - \boldsymbol{Y})) - \nabla f(\boldsymbol{Y}), \boldsymbol{X} - \boldsymbol{Y} \rangle \, dt \\
&\leq f(\boldsymbol{Y}) + \langle \nabla f(\boldsymbol{Y}), \boldsymbol{Y} - \boldsymbol{X} \rangle + \int_{t=0}^{1} \|\nabla f(\boldsymbol{Y} + t(\boldsymbol{X} - \boldsymbol{Y})) - \nabla f(\boldsymbol{Y})\|_{\star} \|\boldsymbol{X} - \boldsymbol{Y}\| \, dt \\
&\leq f(\boldsymbol{Y}) + \langle \nabla f(\boldsymbol{Y}), \boldsymbol{Y} - \boldsymbol{X} \rangle + \int_{t=0}^{1} Lt \, \|\boldsymbol{X} - \boldsymbol{Y}\|^2 \, dt \\
&= f(\boldsymbol{Y}) + \langle \nabla f(\boldsymbol{Y}), \boldsymbol{Y} - \boldsymbol{X} \rangle + \frac{L}{2} \|\boldsymbol{X} - \boldsymbol{Y}\|^2 ,
\end{aligned}
$$

where we use Lemma 3 and Assumption 1 for the first and second inequalities, respectively. ☐

### E.3 MAIN PROOF

**Lemma 5.** *Suppose that both $r = r'$ and $k \geq k'$ hold, or $r > r'$ holds. Then, we have*

$$
\left\| \boldsymbol{X}^{(r,k)} - \boldsymbol{X}^{(r',k')} \right\| \leq \frac{\eta S}{n} ((r - r')K + k - k')
$$

*Proof.* From the update rule of $\boldsymbol{X}^{(r,k)}$ and $\boldsymbol{X}^{(r',k')}$, we have

$$
\begin{aligned}
\boldsymbol{X}^{(r,k)} &= \frac{n - S}{n} \boldsymbol{X}^{(r)} + \frac{1}{n} \sum_{i \in \mathcal{S}_r} \boldsymbol{X}_i^{(r,k)} \\
&= \boldsymbol{X}^{(r)} + \frac{\eta}{n} \sum_{i \in \mathcal{S}_r} \sum_{k''=1}^{k} \boldsymbol{D}_i^{(r,k'')} \\
&= \boldsymbol{X}^{(r')} + \frac{\eta}{n} \sum_{i \in \mathcal{S}_r} \sum_{k''=1}^{k} \boldsymbol{D}_i^{(r,k'')} + \frac{\eta}{n} \sum_{r''=r'}^{r-1} \sum_{i \in \mathcal{S}_{r''}} \sum_{k''=1}^{K} \boldsymbol{D}_i^{(r'',k'')}, \\
\boldsymbol{X}^{(r',k')} &= \boldsymbol{X}^{(r')} + \frac{\eta}{n} \sum_{i \in \mathcal{S}_{r'}} \sum_{k''=1}^{k'} \boldsymbol{D}_i^{(r',k'')}.
\end{aligned}
$$

Thus, we have

$$
\begin{aligned}
\left\| \boldsymbol{X}^{(r,k)} - \boldsymbol{X}^{(r',k')} \right\| &= \left\| \frac{\eta}{n} \sum_{i \in \mathcal{S}_r} \sum_{k''=1}^{k} \boldsymbol{D}_i^{(r,k'')} + \frac{\eta}{n} \sum_{r''=r'+1}^{r-1} \sum_{i \in \mathcal{S}_{r''}} \sum_{k''=1}^{K} \boldsymbol{D}_i^{(r'',k'')} + \frac{\eta}{n} \sum_{i \in \mathcal{S}_{r'}} \sum_{k''=k'+1}^{K} \boldsymbol{D}_i^{(r',k'')} \right\| \\
&\leq \frac{\eta S}{n} ((r - r')K + k - k'),
\end{aligned}
$$

where we use $\|\boldsymbol{D}_i^{(r,k)}\| = 1$ for any $r$ and $k$. ☐

**Lemma 6.** *Suppose that both $r = r'$ and $k \geq k'$ hold, or $r > r'$ holds. Then, we have*

$$
\left\| \boldsymbol{X}_i^{(r,k)} - \boldsymbol{X}_i^{(r',k')} \right\| \leq (r - r' + 2)K\eta.
$$

*Proof.* We have

$$
\begin{aligned}
\left\| \boldsymbol{X}_i^{(r,k)} - \boldsymbol{X}_i^{(r',k')} \right\| &\leq \left\| \boldsymbol{X}_i^{(r,k)} - \boldsymbol{X}_i^{(r,0)} \right\| + \left\| \boldsymbol{X}_i^{(r,0)} - \boldsymbol{X}_i^{(r',0)} \right\| + \left\| \boldsymbol{X}_i^{(r',k')} - \boldsymbol{X}_i^{(r',0)} \right\| \\
&= \left\| \boldsymbol{X}_i^{(r,k)} - \boldsymbol{X}_i^{(r,0)} \right\| + \left\| \boldsymbol{X}^{(r,0)} - \boldsymbol{X}^{(r',0)} \right\| + \left\| \boldsymbol{X}_i^{(r',k')} - \boldsymbol{X}_i^{(r',0)} \right\| \\
&\leq \eta(k + k') + \left\| \boldsymbol{X}^{(r,0)} - \boldsymbol{X}^{(r',0)} \right\|.
\end{aligned}
$$

Using Lemma 5, we obtain the desired result. ☐

**Lemma 7.** *Suppose that Assumptions 1 and 2 hold. Then, when $r \geq 1$, we have*

$$\mathbb{E}f(\boldsymbol{X}^{(r,k+1)}) \leq \mathbb{E}f(\boldsymbol{X}^{(r,k)}) - \frac{\eta S}{n}\left\|\nabla f(\boldsymbol{X}^{(r,k)})\right\|_{\star} + 2LK\left(\frac{S}{n}\right)^2 \eta^2$$
$$+ 2\left(\frac{S}{n}\right)\eta\mathbb{E}\left\|\nabla f(\boldsymbol{X}^{(r-1,K-1)}) - \boldsymbol{C}^{(r)}\right\|_{\star} + \frac{2\eta}{n}\mathbb{E}\sum_{i\in\mathcal{S}_r}\left\|\boldsymbol{M}_i^{(r,k+1)} - \boldsymbol{C}_i^{(r)}\right\|_{\star} + \frac{L}{2}\left(\frac{S}{n}\right)\eta^2.$$

*When $r = 0$, we have*

$$\mathbb{E}f(\boldsymbol{X}^{(0,k+1)}) \leq \mathbb{E}f(\boldsymbol{X}^{(0,k)}) - \frac{\eta S}{n}\left\|\nabla f(\boldsymbol{X}^{(0,k)})\right\|_{\star} + 2LK\left(\frac{S}{n}\right)^2 \eta^2$$
$$+ \frac{2\eta}{n}\mathbb{E}\sum_{i\in\mathcal{S}_0}\left\|\boldsymbol{M}_i^{(0,k+1)} - \boldsymbol{C}_i^{(0)}\right\|_{\star} + \frac{L}{2}\left(\frac{S}{n}\right)\eta^2 + 2\left(\frac{S}{n}\right)\rho\sigma_0\eta.$$

*Proof.* We have

$$\mathbb{E}_{r,k}f(\boldsymbol{X}^{(r,k+1)})$$
$$= \mathbb{E}_{r,k}f\left(\boldsymbol{X}^{(r,k)} + \frac{\eta}{n}\sum_{i\in\mathcal{S}_r}\boldsymbol{D}_i^{(r,k)}\right)$$
$$\leq f(\boldsymbol{X}^{(r,k)}) + \frac{\eta}{n}\mathbb{E}_{r,k}\sum_{i\in\mathcal{S}_r}\left\langle\nabla f(\boldsymbol{X}^{(r,k)}), \boldsymbol{D}_i^{(r,k)}\right\rangle + \frac{L\eta^2}{2n}\mathbb{E}_{r,k}\sum_{i\in\mathcal{S}_r}\left\|\boldsymbol{D}_i^{(r,k+1)}\right\|^2$$
$$\leq f(\boldsymbol{X}^{(r,k)}) + \frac{\eta}{n}\mathbb{E}_{r,k}\sum_{i\in\mathcal{S}_r}\left\langle\nabla f(\boldsymbol{X}^{(r,k)}) - \boldsymbol{G}_i^{(r,k+1)}, \boldsymbol{D}_i^{(r,k+1)}\right\rangle + \frac{\eta}{n}\mathbb{E}_{r,k}\sum_{i\in\mathcal{S}_r}\left\langle\boldsymbol{G}_i^{(r,k+1)}, \boldsymbol{D}_i^{(r,k+1)}\right\rangle + \frac{LS\eta^2}{2n}$$
$$\leq f(\boldsymbol{X}^{(r,k)}) + \frac{\eta}{n}\mathbb{E}_{r,k}\sum_{i\in\mathcal{S}_r}\left\|\nabla f(\boldsymbol{X}^{(r,k)}) - \boldsymbol{G}_i^{(r,k+1)}\right\|_{\star} + \frac{\eta}{n}\mathbb{E}_{r,k}\sum_{i\in\mathcal{S}_r}\underbrace{\left\langle\boldsymbol{G}_i^{(r,k+1)}, \boldsymbol{D}_i^{(r,k+1)}\right\rangle}_{\mathcal{T}_1} + \frac{LS\eta^2}{2n},$$

where we use Lemma 4, $\|\boldsymbol{D}_i^{(r,k+1)}\| \leq 1$, and the Cauchy-Schwarz inequality in the first, second, and third inequalities, and $\boldsymbol{G}_i$ and $\boldsymbol{D}_i$ are defined in Appendix E.1. Using Lemma 1 and the triangle inequality, we have

$$\mathcal{T}_1 = -\left\|\boldsymbol{G}_i^{(r,k+1)}\right\|_{\star} \leq -\left\|\nabla f(\boldsymbol{X}^{(r,k)})\right\|_{\star} + \left\|\nabla f(\boldsymbol{X}^{(r,k)}) - \boldsymbol{G}_i^{(r,k+1)}\right\|_{\star}.$$

Then, it holds

$$\mathbb{E}_{r,k}f(\boldsymbol{X}^{(r,k+1)}) \leq f(\bar{\boldsymbol{X}}^{(r,k)}) - \frac{\eta S}{n}\left\|\nabla f(\boldsymbol{X}^{(r,k)})\right\|_{\star} + \frac{2\eta}{n}\mathbb{E}_{r,k}\sum_{i\in\mathcal{S}_r}\underbrace{\left\|\nabla f(\boldsymbol{X}^{(r,k)}) - \boldsymbol{G}_i^{(r,k+1)}\right\|_{\star}}_{\mathcal{T}_2} + \frac{LS\eta^2}{2n}.$$

When $r \geq 1$, we have

$$\mathcal{T}_2 = \left\|\nabla f(\boldsymbol{X}^{(r,k)}) - \boldsymbol{M}_i^{(r,k+1)} + \boldsymbol{C}_i^{(r)} - \boldsymbol{C}^{(r)}\right\|_{\star}$$
$$\leq \left\|\nabla f(\boldsymbol{X}^{(r,k)}) - \nabla f(\boldsymbol{X}^{(r-1,K-1)})\right\|_{\star} + \left\|\nabla f(\boldsymbol{X}^{(r-1,K-1)}) - \boldsymbol{C}^{(r)}\right\|_{\star} + \left\|\boldsymbol{M}_i^{(r,k+1)} - \boldsymbol{C}_i^{(r)}\right\|_{\star}$$
$$\leq L\left\|\boldsymbol{X}^{(r,k)} - \boldsymbol{X}^{(r-1,K-1)}\right\| + \left\|\nabla f(\boldsymbol{X}^{(r-1,K-1)}) - \boldsymbol{C}^{(r)}\right\|_{\star} + \left\|\boldsymbol{M}_i^{(r,k+1)} - \boldsymbol{C}_i^{(r)}\right\|_{\star}$$
$$\leq \frac{LSK\eta}{n} + \left\|\nabla f(\boldsymbol{X}^{(r-1,K-1)}) - \boldsymbol{C}^{(r)}\right\|_{\star} + \left\|\boldsymbol{M}_i^{(r,k+1)} - \boldsymbol{C}_i^{(r)}\right\|_{\star},$$

where we use Lemma 5 in the last inequality.

When $r = 0$, we have

$$
\begin{aligned}
\mathcal{T}_2 &= \left\| \nabla f(\boldsymbol{X}^{(0,k)}) - \boldsymbol{M}_i^{(0,k+1)} + \boldsymbol{C}_i^{(0)} - \boldsymbol{C}^{(0)} \right\|_\star \\
&\leq \left\| \nabla f(\boldsymbol{X}^{(0,k)}) - \nabla f(\boldsymbol{X}^{(0,0)}) \right\|_\star + \left\| \nabla f(\boldsymbol{X}^{(0,0)}) - \boldsymbol{C}^{(0)} \right\|_\star + \left\| \boldsymbol{M}_i^{(0,k+1)} - \boldsymbol{C}_i^{(0)} \right\|_\star \\
&\leq L \left\| \boldsymbol{X}^{(0,k)} - \boldsymbol{X}^{(0,0)} \right\| + \left\| \nabla f(\boldsymbol{X}^{(0,0)}) - \boldsymbol{C}^{(0)} \right\|_\star + \left\| \boldsymbol{M}_i^{(0,k+1)} - \boldsymbol{C}_i^{(0)} \right\|_\star \\
&\leq \frac{LSK\eta}{n} + \left\| \nabla f(\boldsymbol{X}^{(0,0)}) - \boldsymbol{C}^{(0)} \right\|_\star + \left\| \boldsymbol{M}_i^{(0,k+1)} - \boldsymbol{C}_i^{(0)} \right\|_\star
\end{aligned}
$$

Then, using the following inequality:

$$
\mathbb{E} \left\| \nabla f(\boldsymbol{X}^{(0,0)}) - \boldsymbol{C}^{(0)} \right\|_\star \leq \frac{\rho}{n} \sum_{i=1}^n \sqrt{\mathbb{E} \left\| \nabla f_i(\boldsymbol{X}^{(0,0)}) - \boldsymbol{C}_i^{(0)} \right\|_F^2} \leq \rho \sigma_0,
$$

we obtain the desired result. $\qquad\square$

**Lemma 8.** *Suppose that Assumptions 1 and 2 hold, $\boldsymbol{C}_i^{(0)} := \boldsymbol{M}_i^{(0,0)}$ and $\boldsymbol{C}^{(0)} := \frac{1}{n} \sum_{i=1}^n \boldsymbol{C}_i^{(0)}$,*

$$
\frac{1}{n} \mathbb{E} \sum_{i \in \mathcal{S}_r} \left\| \nabla F_i(\boldsymbol{X}_i^{(r-1,K-1)}) - \boldsymbol{M}_i^{(r,0)} \right\|_\star \leq \frac{S\rho\sigma}{n} \left( 1 - \frac{S\alpha}{n} \right)^{r-1} + \frac{S}{n} \alpha\rho\sqrt{K\sigma^2} + 6LK\eta.
$$

*Proof.* Let $c_i(r-1)$ be the number of times that client $i$ has been sampled by round $r$. We have

$$
\boldsymbol{M}_i^{(r,0)} = (1-\alpha)^{c_i(r-1)K} \boldsymbol{M}_i^{(0,0)} + \alpha \sum_{r'=1}^{c_i(r-1)} \sum_{k'=0}^{K-1} (1-\alpha)^{(c_i(r-1)-r')K+k'} \nabla F_i(\boldsymbol{X}_i^{(r',k')}; \xi_i^{r',k'})
$$

To simplify the notation, we denote $r_i(r')$ by the number of rounds that client $i$ is sampled for the $r'$-th time. Using this notation, we have

$$
\boldsymbol{M}_i^{(r,0)}
$$
$$
= (1-\alpha)^{c_i(r-1)K} \boldsymbol{M}_i^{(0,0)} + \alpha \sum_{c'=1}^{c_i(r-1)K} (1-\alpha)^{c_i(r-1)K-c'} \nabla F_i(\boldsymbol{X}_i^{(r'(\lceil \frac{c'}{K} \rceil), c'-K\lceil \frac{c'}{K} \rceil)}; \xi_i^{(r'(\lceil \frac{c'}{K} \rceil), c'-K\lceil \frac{c'}{K} \rceil)})
$$
$$
= (1-\alpha)^{c_i(r-1)K} \left( \nabla F_i(\boldsymbol{X}_i^{(0,0)}; \xi_i^{(0,0)}) - \nabla f_i(\boldsymbol{X}_i^{(0,0)}) \right)
$$
$$
+ \alpha \sum_{c'=1}^{c_i(r-1)K} (1-\alpha)^{c_i(r-1)K-c'} \left( \nabla F_i(\boldsymbol{X}_i^{(r'(\lceil \frac{c'}{K} \rceil), c'-K\lceil \frac{c'}{K} \rceil)}; \xi_i^{(r'(\lceil \frac{c'}{K} \rceil), c'-K\lceil \frac{c'}{K} \rceil)}) - \nabla f_i(\boldsymbol{X}_i^{(r'(\lceil \frac{c'}{K} \rceil), c'-K\lceil \frac{c'}{K} \rceil)}) \right)
$$
$$
+ \underbrace{(1-\alpha)^{c_i(r-1)K} \nabla f_i(\boldsymbol{X}_i^{(0,0)}) + \alpha \sum_{c'=1}^{c_i(r-1)K} (1-\alpha)^{c_i(r-1)K-c'} \nabla f_i(\boldsymbol{X}_i^{(r'(\lceil \frac{c'}{K} \rceil), c'-L\lceil \frac{c'}{K} \rceil)})}_{\mathcal{T}}.
$$

Using $\alpha(1-\alpha)^m = (1-\alpha)^m - (1-\alpha)^{m+1}$, we have

$$
\mathcal{T} = \nabla f_i(\boldsymbol{X}_i^{(r_i(c_i(r-1)), K-1)})
$$
$$
+ \sum_{c'=1}^{c_i(r-1)K-1} (1-\alpha)^{c_i(r-1)K-c'} \left( \nabla f_i(\boldsymbol{X}_i^{(r'(\lceil \frac{c'}{K} \rceil), c'-L\lceil \frac{c'}{K} \rceil)}) - \nabla f_i(\boldsymbol{X}_i^{(r'(\lceil \frac{c'+1}{K} \rceil), c'+1-K\lceil \frac{c'+1}{K} \rceil)}) \right)
$$
$$
+ (1-\alpha)^{c_i(r-1)K} \left( \nabla f_i(\boldsymbol{X}_i^{(0,,0)}) - \nabla f_i(\boldsymbol{X}_i^{(r_i(1),0)}) \right).
$$

Thus, we have

$$\mathbb{E}\left\|\boldsymbol{M}_i^{(r,0)} - \nabla f_i(\boldsymbol{X}_i^{(r-1,K-1)})\right\|_\star$$

$$\leq \mathbb{E}(1-\alpha)^{c_i(r-1)K}\rho\sigma$$

$$+ \alpha\mathbb{E}\left\|\sum_{c'=1}^{c_i(r-1)K}(1-\alpha)^{c_i(r-1)K-c'}\left(\nabla F_i(\boldsymbol{X}_i^{(r'(\lceil\frac{c'}{K}\rceil),c'-K\lceil\frac{c'}{K}\rceil)};\xi_i^{(r'(\lceil\frac{c'}{K}\rceil),c'-K\lceil\frac{c'}{K}\rceil)}) - \nabla f_i(\boldsymbol{X}_i^{(r'(\lceil\frac{c'}{K}\rceil),c'-K\lceil\frac{c'}{K}\rceil)})\right)\right\|_\star$$

$$+ \mathbb{E}\left\|\nabla f_i(\boldsymbol{X}_i^{(r_i(c_i(r-1)),K-1)}) - \nabla f_i(\boldsymbol{X}_i^{(r-1,K-1)})\right\|_\star$$

$$+ \mathbb{E}\sum_{c'=1}^{c_i(r-1)K-1}(1-\alpha)^{c_i(r-1)K-c'}\left\|\nabla f_i(\boldsymbol{X}_i^{(r'(\lceil\frac{c'}{K}\rceil),c'-L\lceil\frac{c'}{K}\rceil)}) - \nabla f_i(\boldsymbol{X}_i^{(r'(\lceil\frac{c'+1}{K}\rceil),c'+1-K\lceil\frac{c'+1}{K}\rceil)})\right\|_\star$$

$$+ \mathbb{E}(1-\alpha)^{c_i(r-1)K}\left\|\nabla f_i(\boldsymbol{X}_i^{(0,,0)}) - \nabla f_i(\boldsymbol{X}_i^{(r_i(1),0)})\right\|_\star$$

Using Assumption 1, we have

$$\mathbb{E}\left\|\boldsymbol{M}_i^{(r,0)} - \nabla f_i(\boldsymbol{X}_i^{(r-1,K-1)})\right\|_\star$$

$$\leq \underbrace{\mathbb{E}(1-\alpha)^{c_i(r-1)K}\rho\sigma}_{\mathcal{T}_1} + \alpha\rho\sqrt{K\sigma^2}$$

$$+ L\,\mathbb{E}\underbrace{\sum_{c'=1}^{c_i(r-1)K-1}(1-\alpha)^{c_i(r-1)K-c'}\left\|\boldsymbol{X}_i^{(r'(\lceil\frac{c'}{K}\rceil),c'-L\lceil\frac{c'}{K}\rceil)} - \boldsymbol{X}_i^{(r'(\lceil\frac{c'+1}{K}\rceil),c'+1-K\lceil\frac{c'+1}{K}\rceil)}\right\|}_{\mathcal{T}_2}$$

$$+ L\,\mathbb{E}\underbrace{(1-\alpha)^{c_i(r-1)K}\left\|\boldsymbol{X}_i^{(0,,0)} - \boldsymbol{X}_i^{(r_i(1),0)}\right\|}_{\mathcal{T}_3}$$

$$+ L\,\mathbb{E}\underbrace{\left\|\boldsymbol{X}_i^{(r_i(c_i(r-1)),K-1)} - \boldsymbol{X}_i^{(r-1,K-1)}\right\|}_{\mathcal{T}_4}.$$

The quantity of $c_i(r-1)$ is the number of rounds in which client $i$ is sampled, which follows the binomial distribution.

$$\mathcal{T}_1 \leq \rho\sigma\sum_{c'=0}^{r-1}(1-\alpha)^{Kc'}\left(\frac{S}{n}\right)^{c'}\left(1-\frac{S}{n}\right)^{r-1-c'}\binom{r-1}{c'}$$

$$\leq \rho\sigma\sum_{c'=0}^{r-1}\left((1-\alpha)\frac{S}{n}\right)^{c'}\left(1-\frac{S}{n}\right)^{r-1-c'}\binom{r-1}{c'}$$

$$= \rho\sigma\left(1-\frac{S\alpha}{n}\right)^{r-1}.$$

$$\mathcal{T}_2 \leq \eta\mathbb{E}\sum_{c'=1}^{c_i(r-1)K-1}(1-\alpha)^{c_i(r-1)K-c'}\left(r_i\left(\lceil\frac{c'+1}{K}\rceil\right) - r_i\left(\lceil\frac{c'}{K}\rceil\right)\right)$$

$$= \eta\mathbb{E}\sum_{c''=1}^{c_i(r-1)K-1}(1-\alpha)^{c''}\underbrace{\left(r_i\left(\lceil\frac{c'(r-1)K-c''+1}{K}\rceil\right) - r_i\left(\lceil\frac{c'(r-1)K-c''+1}{K}\rceil\right)\right)}_{\mathcal{T}_5}.$$

The quantity of $\mathcal{T}_5$ is the number of rounds from the time cline $i$ was sampled to the next sampling, which follows a geometric distribution with expectation $\frac{n}{S}$. Thus, we have

$$\mathcal{T}_2 \leq \frac{Kn\eta}{S}$$

Using Lemma 6, we have

$$\mathcal{T}_3 \leq \mathbb{E}\left\|\boldsymbol{X}_i^{(0,0)} - \boldsymbol{X}_i^{(r_i(1),0)}\right\|_\star = \mathbb{E}\left\|\boldsymbol{X}^{(0,0)} - \boldsymbol{X}^{(r_i(1),0)}\right\|_\star \leq \frac{\eta S}{n} K \mathbb{E} r_i(1),$$

where we use Lemma 5 in the last inequality. The quantity of $r_i(1)$ is the round in which client $i$ is sampled for the first time, which follows a geometric distribution. Thus, we have

$$\mathcal{T}_3 \leq K\eta.$$

Using Lemma 6, we have

$$\mathcal{T}_4 = K\eta \left( \mathbb{E} \left( r - 1 - r_i(c_i(r-1)) + 2 \right) \right).$$

Since the quantity of $r_i(c_i(r-1))$ is the rounds in which client $i$ is sampled for the last time, we have

$$\mathbb{E}\left(r - 1 - r_i(c_i(r-1))\right) = (r-1)\left(1 - \frac{S}{n}\right)^r + \sum_{r'=0}^{r} r'\left(1 - \frac{S}{n}\right)^{r'} \frac{S}{n} \leq \frac{2n}{S}.$$

Thus, it holds that

$$\mathcal{T}_4 \leq \frac{4Kn\eta}{S}.$$

By combining the above inequalities, we obtain the desired result. $\qquad\square$

**Lemma 9.** *Suppose that Assumptions 1 and 2 holds. When $r \geq 1$, it holds that*

$$\frac{1}{n}\mathbb{E}\sum_{i\in\mathcal{S}_r}\left\|\boldsymbol{M}_i^{(r,k+1)} - \boldsymbol{C}_i^{(r)}\right\|_\star \leq 2\alpha\left(\frac{S}{n}\right)\rho\sqrt{K\sigma^2} + 9KL\eta + \left(\frac{S}{n}\right)\rho\sigma_0\left(1 - \frac{S\alpha}{n}\right)^{r-1},$$

*where $\rho := \sup_{\boldsymbol{X}\in\mathcal{X}} \frac{\|\boldsymbol{X}\|_\star}{\|\boldsymbol{X}\|_F}$ and $\sigma_0^2 := \frac{1}{n}\sum_{i=1}^{n}\mathbb{E}\|\nabla f_i(\boldsymbol{X}_i^{(0)}) - \boldsymbol{C}_i^{(0)}\|_F^2$.*

*Then, when $r = 0$, we have*

$$\frac{1}{n}\mathbb{E}\sum_{i\in\mathcal{S}_0}\left\|\boldsymbol{M}_i^{(0,k+1)} - \boldsymbol{C}_i^{(0)}\right\|_\star \leq \alpha\left(\frac{S}{n}\right)\rho\sqrt{K\sigma^2} + LK\eta + \left(\frac{S}{n}\right)\rho\sigma_0.$$

*Proof.* We have

$$\boldsymbol{M}_i^{(r,k+1)} = (1-\alpha)^{k+1}\boldsymbol{M}_i^{(r,0)} + \alpha\sum_{k'=0}^{k}(1-\alpha)^{k-k'}\nabla F_i(\boldsymbol{X}_i^{(r,k')};\xi_i^{(r,k')}).$$

Since we have $\boldsymbol{C}_i^{(r)} = \boldsymbol{M}_i^{(r,0)}$, we have

$$\mathbb{E}\sum_{i\in\mathcal{S}_r}\left\|\boldsymbol{M}_i^{(r,k+1)} - \boldsymbol{C}_i^{(r)}\right\|_\star = \alpha\mathbb{E}\sum_{i\in\mathcal{S}_r}\left\|\sum_{k'=0}^{k}(1-\alpha)^{k-k'}\left(\nabla F_i(\boldsymbol{X}_i^{(r,k')};\xi_i^{(r,k')}) - \boldsymbol{M}_i^{(r,0)}\right)\right\|_\star.$$

When $r \geq 1$, we have

$$
\begin{aligned}
\mathbb{E} \sum_{i \in \mathcal{S}_r} \left\| \boldsymbol{M}_i^{(r,k+1)} - \boldsymbol{C}_i^{(r)} \right\|_\star &\leq \alpha \mathbb{E} \sum_{i \in \mathcal{S}_r} \left\| \sum_{k'=0}^{k} (1-\alpha)^{k-k'} \left( \nabla F_i(\boldsymbol{X}_i^{(r,k')}; \xi_i^{(r,k')}) - \nabla f_i(\boldsymbol{X}_i^{(r,k')}) \right) \right\|_\star \\
&\quad + \alpha \mathbb{E} \sum_{i \in \mathcal{S}_r} \left\| \sum_{k'=0}^{k} (1-\alpha)^{k-k'} \left( \nabla f_i(\boldsymbol{X}_i^{(r,k')}) - \nabla f_i(\boldsymbol{X}_i^{(r-1,K-1)}) \right) \right\|_\star \\
&\quad + \alpha \mathbb{E} \sum_{i \in \mathcal{S}_r} \left\| \sum_{k'=0}^{k} (1-\alpha)^{k-k'} \left( \nabla f_i(\boldsymbol{X}_i^{(r-1,K-1)}) - \boldsymbol{M}_i^{(r,0)} \right) \right\|_\star \\
&\leq \alpha \mathbb{E} \sum_{i \in \mathcal{S}_r} \left\| \sum_{k'=0}^{k} (1-\alpha)^{k-k'} \left( \nabla F_i(\boldsymbol{X}_i^{(r,k')}; \xi_i^{(r,k')}) - \nabla f_i(\boldsymbol{X}_i^{(r,k')}) \right) \right\|_\star \\
&\quad + \alpha \mathbb{E} \sum_{i \in \mathcal{S}_r} \left\| \sum_{k'=0}^{k} (1-\alpha)^{k-k'} \left( \nabla f_i(\boldsymbol{X}_i^{(r,k')}) - \nabla f_i(\boldsymbol{X}_i^{(r-1,K-1)}) \right) \right\|_\star \\
&\quad + \mathbb{E} \sum_{i \in \mathcal{S}_r} \left\| \nabla f_i(\boldsymbol{X}_i^{(r-1,K-1)}) - \boldsymbol{M}_i^{(r,0)} \right\|_\star .
\end{aligned}
$$

The first term is bounded from above as follows:

$$
\begin{aligned}
\mathbb{E} &\left\| \sum_{k'=0}^{k} (1-\alpha)^{k-k'} \left( \nabla F_i(\boldsymbol{X}_i^{(r,k')}; \xi_i^{(r,k')}) - \nabla f_i(\boldsymbol{X}_i^{(r,k')}) \right) \right\|_\star \\
&\leq \sqrt{ \mathbb{E} \left\| \sum_{k'=0}^{k} (1-\alpha)^{k-k'} \left( \nabla F_i(\boldsymbol{X}_i^{(r,k')}; \xi_i^{(r,k')}) - \nabla f_i(\boldsymbol{X}_i^{(r,k')}) \right) \right\|_\star^2 } \\
&= \sqrt{ \sum_{k'=0}^{k} (1-\alpha)^{2(k-k')} \mathbb{E} \left\| \nabla F_i(\boldsymbol{X}_i^{(r,k')}; \xi_i^{(r,k')}) - \nabla f_i(\boldsymbol{X}_i^{(r,k')}) \right\|_\star^2 } \\
&\leq \rho \sqrt{K \sigma^2},
\end{aligned}
$$

where we used Jensen's inequality in the first inequality. The second term is bounded as follows:

$$
\begin{aligned}
&\left\| \sum_{k'=0}^{k} (1-\alpha)^{k-k'} \left( \nabla f_i(\boldsymbol{X}_i^{(r,k')}) - \nabla f_i(\boldsymbol{X}_i^{(r-1,K-1)}) \right) \right\|_\star \\
&\leq \sum_{k'=0}^{k} (1-\alpha)^{k-k'} \left\| \nabla f_i(\boldsymbol{X}_i^{(r,k')}) - \nabla f_i(\boldsymbol{X}_i^{(r-1,K-1)}) \right\|_\star \\
&\leq L \sum_{k'=0}^{k} (1-\alpha)^{k-k'} \left\| \boldsymbol{X}_i^{(r,k')} - \boldsymbol{X}_i^{(r-1,K-1)} \right\| \\
&\leq \frac{3LK\eta}{\alpha},
\end{aligned}
$$

where we use Lemma 6 in the last inequality. Then, using Lemma 8, we obtain the desired result when $r \geq 1$.

When $r = 0$, we have

$$
\begin{aligned}
\mathbb{E} \sum_{i \in \mathcal{S}_0} \left\| \boldsymbol{M}_i^{(0,k+1)} - \boldsymbol{C}_i^{(0)} \right\|_\star &= \alpha \mathbb{E} \sum_{i \in \mathcal{S}_r} \left\| \sum_{k'=0}^{k} (1-\alpha)^{k-k'} \left( \nabla F_i(\boldsymbol{X}_i^{(r,k')}; \xi_i^{(r,k')}) - \boldsymbol{M}_i^{(r,0)} \right) \right\|_\star \\
&\leq \alpha \mathbb{E} \sum_{i \in \mathcal{S}_0} \left\| \sum_{k'=0}^{k} (1-\alpha)^{k-k'} \left( \nabla F_i(\boldsymbol{X}_i^{(0,k')}; \xi_i^{(0,k')}) - \nabla f_i(\boldsymbol{X}_i^{(0,k')}) \right) \right\|_\star \\
&\quad + \alpha \mathbb{E} \sum_{i \in \mathcal{S}_0} \left\| \sum_{k'=0}^{k} (1-\alpha)^{k-k'} \left( \nabla f_i(\boldsymbol{X}_i^{(0,k')}) - \nabla f_i(\boldsymbol{X}_i^{(0,0)}) \right) \right\|_\star \\
&\quad + \alpha \mathbb{E} \sum_{i \in \mathcal{S}_0} \left\| \sum_{k'=0}^{k} (1-\alpha)^{k-k'} \left( \nabla f_i(\boldsymbol{X}_i^{(0,0)}) - \boldsymbol{M}_i^{(0,0)} \right) \right\|_\star \\
&\leq \alpha S \rho \sqrt{K \sigma^2} + L K S \eta + S \rho \mathbb{E} \left\| \nabla f_i(\boldsymbol{X}_i^{(0,0)}) - \boldsymbol{C}_i^{(0)} \right\|_F,
\end{aligned}
$$

where we use Assumptions 1 and 2 and $\boldsymbol{C}_i^{(0)} = \boldsymbol{M}_i^{(r,0)}$ in the last inequality. Dividing both sides by $n$, we obtain the desired result. $\qquad \square$

**Lemma 10.** *Suppose that Assumptions 1 and 2 hold, $\boldsymbol{C}_i^{(0)} := \boldsymbol{M}_i^{(0,0)}$ and $\boldsymbol{C}^{(0)} := \frac{1}{n} \sum_{i=1}^{n} \boldsymbol{C}_i^{(0)}$, we have*

$$
\mathbb{E} \left\| \nabla f(\boldsymbol{X}^{(r-1,K-1)}) - \boldsymbol{C}^{(r)} \right\|_\star \leq \frac{\rho S}{n} \sqrt{\frac{\alpha \sigma^2}{S}} + \frac{4nL\eta}{\alpha S} + \frac{6LKn\eta}{S} + \rho \sigma \left( 1 - \frac{S\alpha}{n} \right)^{r-1}.
$$

*Proof.* Let $c_i(r-1)$ be the number of times that client $i$ has been sampled by round $r$. We denote $r_i(r')$ by the number of rounds that client $i$ is sampled for the $r'$-th time. Using this notation, we have

$$
\boldsymbol{C}^{(r)} = \frac{1}{n} \sum_{i=1}^{n} \boldsymbol{M}_i^{(r_i(c_i(r-1)),K)}.
$$

Then, we have

$$
\begin{aligned}
\boldsymbol{M}_i^{(r-1,K-1)} &= (1-\alpha)^{c_i(r-1)K} \boldsymbol{M}_i^{(0,0)} + \alpha \sum_{r'=1}^{c_i(r-1)} \sum_{k'=0}^{K-1} (1-\alpha)^{(c_i(r-1)-r')K+k} \nabla F_i(\boldsymbol{X}_i^{(r',k')}; \xi_i^{r',k'}) \\
&= (1-\alpha)^{c_i(r-1)K} \left( \boldsymbol{M}_i^{(0,0)} - \nabla f_i(\boldsymbol{X}^{(0,0)}) \right) \\
&\quad + \underbrace{\alpha \sum_{r'=1}^{c_i(r-1)} \sum_{k'=0}^{K-1} (1-\alpha)^{(c_i(r-1)-r')K+k} \left( \nabla F_i(\boldsymbol{X}_i^{(r',k')}; \xi_i^{r',k'}) - \nabla f_i(\boldsymbol{X}_i^{(r',k')}) \right)}_{\mathcal{T}_1} \\
&\quad + \underbrace{(1-\alpha)^{c_i(r-1)K} \nabla f_i(\boldsymbol{X}^{(0,0)}) + \alpha \sum_{r'=1}^{c_i(r-1)} \sum_{k'=0}^{K-1} (1-\alpha)^{(c_i(r-1)-r')K+k} \nabla f_i(\boldsymbol{X}_i^{(r',k')})}_{\mathcal{T}_2}.
\end{aligned}
$$

We can rewrite $\mathcal{T}_1$ and $\mathcal{T}_2$ as follows:

$$
\mathcal{T}_1 = \alpha \sum_{r'=0}^{r-1} \sum_{k'=1}^{K} \mathbb{1}_{i \in \mathcal{S}_{r'}} (1-\alpha)^{(c_i(r-1)-r'+1)K-k'} \left( \nabla F_i(\boldsymbol{X}_i^{(r',k')}; \xi_i^{(r',k')}) - \nabla f_i(\boldsymbol{X}_i^{(r',k')}) \right).
$$

$$\mathcal{T}_2 = (1-\alpha)^{c_i(r-1)K}\nabla f_i(\boldsymbol{X}^{(0,0)}) + \alpha \sum_{c'=1}^{c_i(r-1)K} (1-\alpha)^{c_i(r-1)K-c'}\nabla f_i(\boldsymbol{X}_i^{(r_i(\lceil \frac{c'}{K}\rceil),c'-K\lceil \frac{c'}{K}\rceil)})$$

$$= \nabla f_i(\boldsymbol{X}_i^{(r_i(c_i(r-1)),K-1)})$$
$$+ \alpha \sum_{c'=1}^{c_i(r-1)K-1} (1-\alpha)^{c_i(r-1)K-c'} \left( \nabla f_i(\boldsymbol{X}_i^{(r_i(\lceil \frac{c'}{K}\rceil),c'-K\lceil \frac{c'}{K}\rceil)}) - \nabla f_i(\boldsymbol{X}_i^{(r_i(\lceil \frac{c'+1}{K}\rceil),c'+1-K\lceil \frac{c'+1}{K}\rceil)}) \right)$$
$$+ (1-\alpha)^{c_i(r-1)K} \left( \nabla f_i(\boldsymbol{X}^{(0,0)}) - \nabla f_i(\boldsymbol{X}_i^{(r_i(1),0)}) \right).$$

Thus, we have

$$\mathbb{E}\left\|\boldsymbol{M}^{(r-1,K-1)} - \nabla f(\boldsymbol{X}^{(r-1,K-1)})\right\|_\star$$

$$\leq \underbrace{\mathbb{E}(1-\alpha)^{c_i(r-1)K}\left\|\frac{1}{n}\sum_{i=1}^n \left(\nabla f_i(\boldsymbol{X}^{(0,0)};\xi_i^{(0,0)}) - \nabla f_i(\boldsymbol{X}^{(0,0)})\right)\right\|_\star}_{\mathcal{T}_3}$$

$$+ \alpha \underbrace{\mathbb{E}\left\|\frac{1}{n}\sum_{i=1}^n\sum_{r'=0}^{r-1}\sum_{k'=1}^K \mathbb{1}_{i\in\mathcal{S}_{r'}}(1-\alpha)^{(c_i(r-1)-r'+1)K-k'} \left(\nabla F_i(\boldsymbol{X}_i^{(r',k')};\xi_i^{(r',k')}) - \nabla f_i(\boldsymbol{X}_i^{(r',k')})\right)\right\|_\star}_{\mathcal{T}_4}$$

$$+ \frac{1}{n}\sum_{i=1}^n \underbrace{\mathbb{E}\left\|\nabla f_i(\boldsymbol{X}_i^{(r_i(c_i(r-1)),K-1)}) - \nabla f_i(\boldsymbol{X}^{(r-1,K-1)})\right\|_\star}_{\mathcal{T}_5}$$

$$+ \frac{1}{n}\sum_{i=1}^n \underbrace{\mathbb{E}\sum_{c'=1}^{c_i(r-1)K-1} (1-\alpha)^{c_i(r-1)K-c'}\left\|\nabla f_i(\boldsymbol{X}_i^{(r_i(\lceil \frac{c'}{K}\rceil),c'-K\lceil \frac{c'}{K}\rceil)}) - \nabla f_i(\boldsymbol{X}_i^{(r_i(\lceil \frac{c'+1}{K}\rceil),c'+1-K\lceil \frac{c'+1}{K}\rceil)})\right\|_\star}_{\mathcal{T}_6}$$

$$+ \frac{1}{n}\sum_{i=1}^n \underbrace{\mathbb{E}(1-\alpha)^{c_i(r-1)K}\left\|\nabla f_i(\boldsymbol{X}^{(0,0)}) - \nabla f_i(\boldsymbol{X}_i^{(r_i(1),0)})\right\|_\star}_{\mathcal{T}_7}.$$

$$\mathcal{T}_3 \leq \mathbb{E}(1-\alpha)^{c_i(r-1)K}\rho\sigma$$

The quantity of $c_i(r-1)$ is the number of rounds in which client $i$ is sampled, which follows the binomial distribution.

$$\mathcal{T}_3 \leq \rho\sigma\sum_{c'=0}^{r-1}(1-\alpha)^{Kc'}\left(\frac{S}{n}\right)^{c'}\left(1-\frac{S}{n}\right)^{r-1-c'}\binom{r-1}{c'}$$

$$\leq \rho\sigma\sum_{c'=0}^{r-1}\left((1-\alpha)\frac{S}{n}\right)^{c'}\left(1-\frac{S}{n}\right)^{r-1-c'}\binom{r-1}{c'}$$

$$= \rho\sigma\left(1-\frac{S\alpha}{n}\right)^{r-1}$$

$$\mathcal{T}_4 = \frac{1}{n}\mathbb{E}\left\|\sum_{r'=0}^{r-1}\sum_{k'=1}^K\sum_{i\in\mathcal{S}_{r'}}(1-\alpha)^{(c_i(r-1)-r'+1)K-k'} \left(\nabla F_i(\boldsymbol{X}_i^{(r',k')};\xi_i^{(r',k')}) - \nabla f_i(\boldsymbol{X}_i^{(r',k')})\right)\right\|_\star$$

$$\leq \frac{\rho S}{n}\sqrt{\frac{\sigma^2}{S(1-(1-\alpha)^2)}}.$$

$$\mathcal{T}_5 = \mathbb{E} \left\| \nabla f_i(\boldsymbol{X}_i^{(r_i(c_i(r-1)),K-1)}) - \nabla f_i(\boldsymbol{X}^{(r-1,K-1)}) \right\|_\star$$

$$\leq L\mathbb{E} \left\| \boldsymbol{X}_i^{(r_i(c_i(r-1)),K-1)} - \boldsymbol{X}^{(r-1,K-1)} \right\|$$

$$\leq L\mathbb{E} \left\| \boldsymbol{X}_i^{(r_i(c_i(r-1)),K-1)} - \boldsymbol{X}_i^{(r_i(c_i(r-1)),0)} \right\| + L\mathbb{E} \left\| \boldsymbol{X}^{(r_i(c_i(r-1)),0)} - \boldsymbol{X}^{(r-1,K-1)} \right\|$$

$$\leq L\eta(K-1) + \frac{LS\eta}{n}\mathbb{E}(((r-1-r_i(c_i(r-1)))\,K + K - 1).$$

Since the quantity of $r_i(c_i(r-1))$ is the rounds in which client $i$ is sampled for the last time, we have

$$\mathbb{E}\left(r-1-r_i(c_i(r-1))\right) = (r-1)\left(1-\frac{S}{n}\right)^r + \sum_{r'=0}^{r} r'\left(1-\frac{S}{n}\right)^{r'}\frac{S}{n} \leq \frac{2n}{S}.$$

Thus, we have

$$\mathcal{T}_5 \leq 2LK\eta.$$

$$\mathcal{T}_6 \leq L\mathbb{E} \sum_{c'=1}^{c_i(r-1)K-1} (1-\alpha)^{c_i(r-1)K-c'} \left\| \boldsymbol{X}_i^{(r_i(\lceil \frac{c'}{K}\rceil),c'-K\lceil\frac{c'}{K}\rceil)} - \boldsymbol{X}_i^{(r_i(\lceil\frac{c'+1}{K}\rceil),c'+1-K\lceil\frac{c'+1}{K}\rceil)} \right\|$$

$$= L\mathbb{E} \sum_{c''=1}^{c_i(r-1)} \sum_{k'=0}^{K-2} (1-\alpha)^{(c_i(r-1)-c'')K-k'} \left\| \boldsymbol{X}_i^{(r_i(c''),k'+1)} - \boldsymbol{X}_i^{(r_i(c''),k')} \right\|$$

$$+ L\mathbb{E} \sum_{c''=1}^{c_i(r-1)} (1-\alpha)^{(c_i(r-1)-c'')K+1} \left\| \boldsymbol{X}_i^{(r_i(c''-1),0)} - \boldsymbol{X}_i^{(r_i(c''),K-1)} \right\|$$

$$\leq +L\mathbb{E} \sum_{c''=1}^{c_i(r-1)} \sum_{k'=0}^{K-2} (1-\alpha)^{(c_i(r-1)-c'')K-k'}$$

$$+ L\eta\mathbb{E} \sum_{c''=1}^{c_i(r-1)-1} (1-\alpha)^{(c_i(r-1)-c'')K+1} \left(r_i(c''+1) - r_i(c'') + 2\right)$$

$$\leq \frac{L\eta}{\alpha} + L\eta\mathbb{E} \sum_{c''=1}^{c_i(r-1)-1} (1-\alpha)^{(c_i(r-1)-c'')K+1} \left(r_i(c''+1) - r_i(c'') + 2\right).$$

The quantity $r_i(c''+1) - r_i(c'')$ follows the geometric distribution, which has the expectation of $\frac{n}{S}$. Using Lemma 2, we obtain

$$\mathcal{T}_6 \leq \frac{4nL\eta}{\alpha S} + \frac{3nLK\eta}{S}.$$

$$\mathcal{T}_7 \leq L\mathbb{E}(1-\alpha)^{c_i(r-1)K} \left\| \boldsymbol{X}_i^{(0,0)} - \boldsymbol{X}_i^{(r_i(1),0)} \right\| \leq LK\eta\mathbb{E}(r_i(1)+2),$$

where we used Lemma 6 in the last inequality. The quantity of $r_i(1)$ is the round in which client $i$ is sampled for the first time, which follows a geometric distribution. Thus, we have

$$\mathcal{T}_7 \leq \frac{3nLK\eta}{S}$$

$$\square$$

**Lemma 11.** *Suppose that Assumptions 1 and 2 hold, $\boldsymbol{C}_i^{(0)} := \boldsymbol{M}_i^{(0,0)}$ and $\boldsymbol{C}^{(0)} := \frac{1}{n}\sum_{i=1}^n \boldsymbol{C}_i^{(0)}$, there exists $\eta$ and $\alpha$ such that we have*

$$\frac{1}{RK}\sum_{r=0}^{R-1}\sum_{k=0}^{K-1}\mathbb{E}\left\|\nabla f(\boldsymbol{X}^{(r,k)})\right\|_\star \leq \mathcal{O}\left(\left(\frac{Lr_0\rho^2\sigma^2}{SRK}\right)^{\frac{1}{4}} + \left(\left(\frac{n}{S}\right)^2\frac{Lr_0\rho\sigma}{R\sqrt{K}}\right)^{\frac{1}{3}} + \left(\frac{Lr_0}{R}\left(\frac{n}{S}\right)^2\right)^{\frac{1}{2}} + \frac{\rho\sigma_0}{R}\left(\frac{n}{S}\right)\right.$$

$$\left.+\rho\sigma_0\left(\frac{\rho^2\sigma^2KS}{Lr_0Rn^2}\right)^{\frac{1}{2}} + \rho\sigma_0\left(\left(\frac{n}{S}\right)\frac{\rho^2\sigma^2K^2}{Lr_0R^2}\right)^{\frac{1}{3}}\right)$$

*Proof.* Combining Lemmas 7, 9 and 10, when $r \geq 1$, it holds

$$\mathbb{E}f(\boldsymbol{X}^{(r,k+1)}) \leq \mathbb{E}f(\boldsymbol{X}^{(r,k)}) - \eta\left(\frac{S}{n}\right)\left\|\nabla f(\boldsymbol{X}^{(r,k)})\right\|_{\star}$$

$$+ 2\rho\eta\left(\frac{S}{n}\right)^2\sqrt{\frac{\alpha\sigma^2}{S}} + \frac{8L\eta^2}{\alpha} + 33LK\eta^2 + 4\alpha\rho\eta\left(\frac{S}{n}\right)\sqrt{K\sigma^2}$$

$$+ 4\rho\eta\left(\frac{S}{n}\right)\sigma_0\left(1 - \frac{S\alpha}{n}\right)^{r-1}.$$

When $r = 0$, we have

$$\mathbb{E}f(\boldsymbol{X}^{(0,k+1)}) \leq \mathbb{E}f(\boldsymbol{X}^{(0,k)}) - \frac{\eta S}{n}\left\|\nabla f(\boldsymbol{X}^{(0,k)})\right\|_{\star}$$

$$+ 5LK\eta^2 + 2\alpha\rho\eta\left(\frac{S}{n}\right)\sqrt{K\sigma^2} + 4\rho\eta\left(\frac{S}{n}\right)\sigma_0.$$

Summing up the above two inequalities, we obtain

$$\frac{1}{RK}\sum_{r=0}^{R-1}\sum_{k=0}^{K-1}\mathbb{E}\left\|\nabla f(\boldsymbol{X}^{(r,k)})\right\|_{\star} \leq \left(\frac{n}{S}\right)\frac{r_0}{RK\eta} + 2\rho\left(\frac{S}{n}\right)\sqrt{\frac{\alpha\sigma^2}{S}} + \frac{8L\eta}{\alpha}\left(\frac{n}{S}\right) + 33LK\left(\frac{n}{S}\right)\eta$$

$$+ 4\alpha\rho\sqrt{K\sigma^2} + \frac{4\rho\sigma_0}{R}\sum_{r=1}^{R-1}\left(1 - \frac{S\alpha}{n}\right)^{r-1} + \frac{4\rho\sigma_0}{R}$$

$$\leq \left(\frac{n}{S}\right)\frac{r_0}{RK\eta} + 2\rho\left(\frac{S}{n}\right)\sqrt{\frac{\alpha\sigma^2}{S}} + \frac{8L\eta}{\alpha}\left(\frac{n}{S}\right) + 33LK\left(\frac{n}{S}\right)\eta$$

$$+ 4\alpha\rho\sqrt{K\sigma^2} + \frac{8\rho\sigma_0}{R\alpha}\left(\frac{n}{S}\right).$$

Then, using the following hyperparameters

$$\eta = \min\left\{\sqrt{\frac{\alpha r_0}{8LRK}}, \frac{1}{K}\sqrt{\frac{r_0}{33LR}}\right\},$$

$$\alpha = \min\left\{1, \left(\frac{n}{S}\right)^2\sqrt{\frac{8Lr_0 S}{RL\rho^2\sigma^2}}, \left(\frac{2Lr_0}{RK^2\rho^2\sigma^2}\left(\frac{n}{S}\right)^2\right)^{\frac{1}{3}}\right\},$$

we obtain the desired result. $\qquad\square$

## F   PROOF OF THEOREM 3

**Lemma 12.** *Let $\boldsymbol{G}$ and $-\boldsymbol{G}^{(T)}$ be the input and output of Algorithm 2 with $a = \frac{15}{8}, b = -\frac{5}{4}$, and $c = \frac{3}{8}$. For any number of iterations $T$, we have*

$$\langle \boldsymbol{G}, -\boldsymbol{G}^{(T)} \rangle \leq -\|\boldsymbol{G}\|_p,$$

*where $p$ is defined as follows:*

$$p := 1 + \frac{\log \left( 1 - (1 - \kappa)^{1.5^T} \right)}{\log \kappa},$$

$$\kappa := \min_i \frac{s_i}{\sqrt{\sum_j s_j^2}} \quad (> 0),$$

*and $s_i$ is the non-zero singular value of $\boldsymbol{G}$.*

*Proof.* Let the singular value decomposition of $\boldsymbol{G}$ be $\boldsymbol{U}\Sigma\boldsymbol{V}$. Then, the output $-\boldsymbol{G}^{(T)}$ can be written as follows:

$$-\boldsymbol{G}^{(T)} = -\boldsymbol{U}\Sigma^{(T)}\boldsymbol{V},$$

where $\Sigma^{(T)}$ is defined as follows:

$$\Sigma_{ii}^{(T)} := \underbrace{\phi\left(\phi\left(\cdots\phi\left(\frac{\Sigma_{ii}}{\|\boldsymbol{G}\|_F}\right)\right)\right)}_{T\text{times}}.$$

Since $\phi(x) > x$, we have

$$\Sigma_{ii}^{(T)} \geq \frac{\Sigma_{ii}}{\|\boldsymbol{G}\|_F}.$$

Using the above inequality, we have

$$\langle \boldsymbol{G}, -\boldsymbol{G}^{(T)} \rangle = -\left\langle \boldsymbol{U}\Sigma\boldsymbol{V}, \boldsymbol{U}\Sigma^{(T)}\boldsymbol{V} \right\rangle$$

$$= -\sum_i \Sigma_{ii}\left(1 - \left(1 - \Sigma_{ii}^{(T)}\right)\right).$$

When $a = \frac{15}{8}, b = -\frac{5}{4}$ and $c = \frac{3}{8}$, we have

$$0 \leq 1 - \phi(x) = (1-x)^2\left(-\frac{3}{8}x^3 - \frac{3}{4}x^2 + \frac{1}{8}x + 1\right) \leq (1-x)^{1.5}$$

Thus, it holds that

$$1 - \Sigma_{ii}^{(T)} \leq \left(1 - \frac{\Sigma_{ii}}{\|\boldsymbol{G}\|_F}\right)^{1.5^T} \leq \left(1 - \frac{\Sigma_{ii}}{\left(\sum_j \Sigma_{jj}^p\right)^{\frac{1}{p}}}\right)^{1.5^T},$$

for any $1 \leq p \leq 2$. Using the above inequality and the definition of $p$ and $\kappa$ we get

$$\langle \boldsymbol{G}, -\boldsymbol{G}^{(T)} \rangle \leq -\left(\sum_i \Sigma_{ii}^p\right)^{\frac{1}{p}}.$$

$\square$

**Lemma 13.** *Suppose that Assumptions 1 and 2 hold. Then, when $r \geq 1$, we have*

$$\mathbb{E}f(\boldsymbol{X}^{(r,k+1)}) \leq \mathbb{E}f(\boldsymbol{X}^{(r,k)}) - \frac{\eta S}{n}\left\|\nabla f(\boldsymbol{X}^{(r,k)})\right\|_p + 2LK\left(\frac{S}{n}\right)^2\eta^2$$

$$+ 2\left(\frac{S}{n}\right)\eta\mathbb{E}\left\|\nabla f(\boldsymbol{X}^{(r-1,K-1)}) - \boldsymbol{C}^{(r)}\right\|_{trace} + \frac{2\eta}{n}\mathbb{E}\sum_{i\in\mathcal{S}_r}\left\|\boldsymbol{M}_i^{(r,k+1)} - \boldsymbol{C}_i^{(r)}\right\|_{trace} + \frac{L}{2}\left(\frac{S}{n}\right)\eta^2,$$

*where $p$ is defined as follows:*

$$p := 1 + \frac{\log\left(1 - (1-\kappa)^{1.5^T}\right)}{\log \kappa},$$

$$\kappa := \min_{j,i,r,k} \frac{s_{j,i,r,k}}{\sqrt{\sum_{j'} s_{j',i,r,k}^2}} \ (> 0),$$

*and $\{s_{j,r,k}\}_j$ are non-zero singular values of $\boldsymbol{M}_i^{(r,k+1)} - \boldsymbol{C}_i^{(r)} + \boldsymbol{C}^{(r)}$.*

*When $r = 0$, we have*

$$\mathbb{E}f(\boldsymbol{X}^{(0,k+1)}) \le \mathbb{E}f(\boldsymbol{X}^{(0,k)}) - \frac{\eta S}{n}\left\|\nabla f(\boldsymbol{X}^{(0,k)})\right\|_p + 2LK\left(\frac{S}{n}\right)^2\eta^2$$

$$+ \frac{2\eta}{n}\mathbb{E}\sum_{i\in\mathcal{S}_0}\left\|\boldsymbol{M}_i^{(0,k+1)} - \boldsymbol{C}_i^{(0)}\right\|_{trace} + \frac{L}{2}\left(\frac{S}{n}\right)\eta^2 + 2\left(\frac{S}{n}\right)\rho\sigma_0\eta.$$

*Proof.* We have

$$\mathbb{E}_{r,k}f(\boldsymbol{X}^{(r,k+1)})$$

$$= \mathbb{E}_{r,k}f\left(\boldsymbol{X}^{(r,k)} + \frac{\eta}{n}\sum_{i\in\mathcal{S}_r}\boldsymbol{D}_i^{(r,k)}\right)$$

$$\le f(\boldsymbol{X}^{(r,k)}) + \frac{\eta}{n}\mathbb{E}_{r,k}\sum_{i\in\mathcal{S}_r}\left\langle\nabla f(\boldsymbol{X}^{(r,k)}),\boldsymbol{D}_i^{(r,k)}\right\rangle + \frac{L\eta^2}{2n}\mathbb{E}_{r,k}\sum_{i\in\mathcal{S}_r}\left\|\boldsymbol{D}_i^{(r,k+1)}\right\|_{sp}^2$$

$$\le f(\boldsymbol{X}^{(r,k)}) + \frac{\eta}{n}\mathbb{E}_{r,k}\sum_{i\in\mathcal{S}_r}\left\langle\nabla f(\boldsymbol{X}^{(r,k)}) - \boldsymbol{G}_i^{(r,k+1)},\boldsymbol{D}_i^{(r,k+1)}\right\rangle + \frac{\eta}{n}\mathbb{E}_{r,k}\sum_{i\in\mathcal{S}_r}\left\langle\boldsymbol{G}_i^{(r,k+1)},\boldsymbol{D}_i^{(r,k+1)}\right\rangle + \frac{LS\eta^2}{2n}$$

$$\le f(\boldsymbol{X}^{(r,k)}) + \frac{\eta}{n}\mathbb{E}_{r,k}\sum_{i\in\mathcal{S}_r}\left\|\nabla f(\boldsymbol{X}^{(r,k)}) - \boldsymbol{G}_i^{(r,k+1)}\right\|_{trace} + \frac{\eta}{n}\mathbb{E}_{r,k}\sum_{i\in\mathcal{S}_r}\underbrace{\left\langle\boldsymbol{G}_i^{(r,k+1)},\boldsymbol{D}_i^{(r,k+1)}\right\rangle}_{\mathcal{T}_1} + \frac{LS\eta^2}{2n},$$

where we use Lemma 4, $\|\boldsymbol{D}_i^{(r,k+1)}\|_{sp} \le 1$, and the Lemma 3 in the first, second, and third inequalities. Using Lemma 12, the definition of $p$, and the triangle inequality, we have

$$\mathcal{T}_1 \le -\left\|\boldsymbol{G}_i^{(r,k+1)}\right\|_p$$

$$\le -\left\|\nabla f(\boldsymbol{X}^{(r,k)})\right\|_p + \left\|\nabla f(\boldsymbol{X}^{(r,k)}) - \boldsymbol{G}_i^{(r,k+1)}\right\|_p$$

$$\le -\left\|\nabla f(\boldsymbol{X}^{(r,k)})\right\|_p + \left\|\nabla f(\boldsymbol{X}^{(r,k)}) - \boldsymbol{G}_i^{(r,k+1)}\right\|_{trace},$$

where we use the fact that $p \ge 1$ and $\|\boldsymbol{A}\|_p \le \|\boldsymbol{A}\|_{trace}$ for any $\boldsymbol{A}$. Then, it holds

$$\mathbb{E}_{r,k}f(\boldsymbol{X}^{(r,k+1)}) \le f(\bar{\boldsymbol{X}}^{(r,k)}) - \frac{\eta S}{n}\left\|\nabla f(\boldsymbol{X}^{(r,k)})\right\|_p + \frac{2\eta}{n}\mathbb{E}_{r,k}\sum_{i\in\mathcal{S}_r}\underbrace{\left\|\nabla f(\boldsymbol{X}^{(r,k)}) - \boldsymbol{G}_i^{(r,k+1)}\right\|_{trace}}_{\mathcal{T}_2} + \frac{LS\eta^2}{2n}.$$

When $r \ge 1$, we have

$$\mathcal{T}_2 = \left\|\nabla f(\boldsymbol{X}^{(r,k)}) - \boldsymbol{M}_i^{(r,k+1)} + \boldsymbol{C}_i^{(r)} - \boldsymbol{C}^{(r)}\right\|_{trace}$$

$$\le \left\|\nabla f(\boldsymbol{X}^{(r,k)}) - \nabla f(\boldsymbol{X}^{(r-1,K-1)})\right\|_{trace} + \left\|\nabla f(\boldsymbol{X}^{(r-1,K-1)}) - \boldsymbol{C}^{(r)}\right\|_{trace} + \left\|\boldsymbol{M}_i^{(r,k+1)} - \boldsymbol{C}_i^{(r)}\right\|_{trace}$$

$$\le L\left\|\boldsymbol{X}^{(r,k)} - \boldsymbol{X}^{(r-1,K-1)}\right\|_{sp} + \left\|\nabla f(\boldsymbol{X}^{(r-1,K-1)}) - \boldsymbol{C}^{(r)}\right\|_{trace} + \left\|\boldsymbol{M}_i^{(r,k+1)} - \boldsymbol{C}_i^{(r)}\right\|_{trace}$$

$$\le \frac{LSK\eta}{n} + \left\|\nabla f(\boldsymbol{X}^{(r-1,K-1)}) - \boldsymbol{C}^{(r)}\right\|_{trace} + \left\|\boldsymbol{M}_i^{(r,k+1)} - \boldsymbol{C}_i^{(r)}\right\|_{trace},$$

where we use Lemma 5 in the last inequality.

When $r = 0$, we have

$$
\begin{aligned}
\mathcal{T}_2 &= \left\| \nabla f(\boldsymbol{X}^{(0,k)}) - \boldsymbol{M}_i^{(0,k+1)} + \boldsymbol{C}_i^{(0)} - \boldsymbol{C}^{(0)} \right\|_{\text{trace}} \\
&\leq \left\| \nabla f(\boldsymbol{X}^{(0,k)}) - \nabla f(\boldsymbol{X}^{(0,0)}) \right\|_{\text{trace}} + \left\| \nabla f(\boldsymbol{X}^{(0,0)}) - \boldsymbol{C}^{(0)} \right\|_{\text{trace}} + \left\| \boldsymbol{M}_i^{(0,k+1)} - \boldsymbol{C}_i^{(0)} \right\|_{\text{trace}} \\
&\leq L \left\| \boldsymbol{X}^{(0,k)} - \boldsymbol{X}^{(0,0)} \right\|_{\text{sp}} + \left\| \nabla f(\boldsymbol{X}^{(0,0)}) - \boldsymbol{C}^{(0)} \right\|_{\text{trace}} + \left\| \boldsymbol{M}_i^{(0,k+1)} - \boldsymbol{C}_i^{(0)} \right\|_{\text{trace}} \\
&\leq \frac{LSK\eta}{n} + \left\| \nabla f(\boldsymbol{X}^{(0,0)}) - \boldsymbol{C}^{(0)} \right\|_{\text{trace}} + \left\| \boldsymbol{M}_i^{(0,k+1)} - \boldsymbol{C}_i^{(0)} \right\|_{\text{trace}}.
\end{aligned}
$$

Then, using the following inequality:

$$
\mathbb{E} \left\| \nabla f(\boldsymbol{X}^{(0,0)}) - \boldsymbol{C}^{(0)} \right\|_{\star} \leq \frac{\rho}{n} \sum_{i=1}^{n} \sqrt{ \mathbb{E} \left\| \nabla f_i(\boldsymbol{X}^{(0,0)}) - \boldsymbol{C}_i^{(0)} \right\|_F^2 } \leq \rho \sigma_0,
$$

we obtain the desired result. $\qquad\square$

**Lemma 14.** *Suppose that Assumptions 1 and 2 hold, $\boldsymbol{C}_i^{(0)} := \boldsymbol{M}_i^{(0,0)}$ and $\boldsymbol{C}^{(0)} := \frac{1}{n} \sum_{i=1}^{n} \boldsymbol{C}_i^{(0)}$, there exists $\eta$ and $\alpha$ such that we have*

$$
\begin{aligned}
\frac{1}{RK} \sum_{r=0}^{R-1} \sum_{k=0}^{K-1} \mathbb{E} \left\| \nabla f(\boldsymbol{X}^{(r,k)}) \right\|_p &\leq \mathcal{O}\left( \left( \frac{Lr_0 \rho^2 \sigma^2}{SRK} \right)^{\frac{1}{4}} + \left( \left( \frac{n}{S} \right)^2 \frac{Lr_0 \rho \sigma}{R\sqrt{K}} \right)^{\frac{1}{3}} + \left( \frac{Lr_0}{R} \left( \frac{n}{S} \right)^2 \right)^{\frac{1}{2}} + \frac{\rho \sigma_0}{R} \left( \frac{n}{S} \right) \right. \\
&\quad \left. + \rho \sigma_0 \left( \frac{\rho^2 \sigma^2 KS}{Lr_0 R n^2} \right)^{\frac{1}{2}} + \rho \sigma_0 \left( \left( \frac{n}{S} \right) \frac{\rho^2 \sigma^2 K^2}{Lr_0 R^2} \right)^{\frac{1}{3}} \right),
\end{aligned}
$$

*Then, $p$ is defined as follows:*

$$
p := 1 + \frac{\log \left( 1 - (1 - \kappa)^{1.5^T} \right)}{\log \kappa},
$$

$$
\kappa := \min_{j,i,r,k} \frac{s_{j,i,r,k}}{\sqrt{\sum_{j'} s_{j',i,r,k}^2}} \quad (> 0),
$$

*where $\{s_{j,i,r,k}\}_j$ are non-zero singular values of $\boldsymbol{M}_i^{(r,k+1)} - \boldsymbol{C}_i^{(r)} + \boldsymbol{C}^{(r)}$.*

*Proof.* Even if we solve the LMO approximately, the statements of Lemmas 9 and 10 hold. Thus, combining Lemmas 9, 10 and 13 and tuning the hyperparameters as in Lemma 11, we obtain the desired result. $\qquad\square$

## G  HYPERPARAMETER TUNING STRATEGY

In our experiments, the hyperparameters were tuned individually for each combination of method, dataset, and random seed.

Table 1: Hyperparameter tuning strategy for each method.

| | | |
|---|---|---|
| FedAvg | Stepsize | Grid search over $\{0.1, 0.01, 0.001\}$ |
| FedAvg (Adam) | Stepsize | Grid search over $\{0.1, 0.01, 0.001\}$ |
| SCAFFOLD | Stepsize | Grid search over $\{0.1, 0.01, 0.001\}$ |
| SCAFFOLD (Adam) | Stepsize | Grid search over $\{0.1, 0.01, 0.001\}$ |
| LocalMuon | Stepsize of Muon | Grid search over $\{0.001, 0.0001\}$ |
| | Stepsize of Momentum SGD | Grid search over $\{0.1, 0.01\}$ |
| FedMuon | Stepsize of Muon | Grid search over $\{0.001, 0.0001\}$ |
| | Stepsize of Momentum SGD | Grid search over $\{0.1, 0.01\}$ |

In the following tables, we list the hyperparameters tuned by the grid search. The reported hyperparameters correspond to the values selected from two independent trials with different random seeds.

Table 2: Hyperparameters tuned for FashionMNIST.

| | $\beta = 10.0$ | $\beta = 0.1$ |
|---|---|---|
| FedAvg | $\{0.1, 0.1\}$ | $\{0.1, 0.1\}$ |
| FedAvg (Adam) | $\{0.1, 0.1\}$ | $\{0.01, 0.01\}$ |
| SCAFFOLD | $\{0.1, 0.1\}$ | $\{0.1, 0.1\}$ |
| SCAFFOLD (Adam) | $\{0.001, 0.001\}$ | $\{0.001, 0.001\}$ |
| LocalMuon | $\{(0.001, 0.1), (0.001, 0.01)\}$ | $\{(0.001, 0.1), (0.001, 0.1)\}$ |
| FedMuon | $\{(0.001, 0.01), (0.001, 0.01)\}$ | $\{(0.001, 0.01), (0.001, 0.01)\}$ |

Table 3: Hyperparameters tuned for CIFAR-10.

| | $\beta = 10.0$ | $\beta = 0.1$ |
|---|---|---|
| FedAvg | $\{0.1, 0.1\}$ | $\{0.1, 0.1\}$ |
| FedAvg (Adam) | $\{0.001, 0.001\}$ | $\{0.01, 0.001\}$ |
| SCAFFOLD | $\{0.1, 0.1\}$ | $\{0.1, 0.1\}$ |
| SCAFFOLD (Adam) | $\{0.001, 0.001\}$ | $\{0.001, 0.001\}$ |
| LocalMuon | $\{(0.001, 0.01), (0.001, 0.01)\}$ | $\{(0.001, 0.01), (0.001, 0.01)\}$ |
| FedMuon | $\{(0.001, 0.01), (0.001, 0.01)\}$ | $\{(0.001, 0.01), (0.001, 0.01)\}$ |

