# OpenReview forum: "FedMuon: Federated Learning with Bias-corrected LMO-based Optimization"
_ICLR.cc/2026/Conference — ICLR 2026 Poster_

### Official Review · Reviewer_YvBN · 2025-10-20

**Soundness:** 2
**Presentation:** 2
**Contribution:** 3
**Rating:** 4
**Confidence:** 4

**Summary:**

This paper studies how to adapt the recently proposed Muon optimizer, an LMO-based method, to the federated learning setting. The authors show that directly applying Muon as a local optimizer fails to converge due to the inherent bias of the LMO operator. To address this, they propose FedMuon, which introduces bias correction via control variates similar to SCAFFOLD. The paper provides a detailed convergence analysis under both exact and approximate LMOs, showing that FedMuon converges for any number of Newton–Schulz iterations, and converges faster when the LMO is solved more accurately. Experiments on FashionMNIST and CIFAR-10 demonstrate superior performance in both homogeneous and heterogeneous settings.

**Strengths:**

1. The paper provides the first formal analysis of bias introduced by LMO in the federated setting and rigorously proves that straightforwardly applying Muon fails to converge.
2. The author establish the convergence under both exact and inexact LMOs, and clarifying how the number of Newton–Schulz iterations affects the rate.
3. The experiments align well with theoretical findings.

**Weaknesses:**

1. The algorithm section (Section 4) only offers an intuitive, SCAFFOLD-like rationale for bias mitigation, but lacks formal theoretical support. It would be helpful to include a short proof sketch or lemma explicitly showing which terms or steps contribute to bias reduction.
2. Appendix D includes an example to prove why LocalMuon does not converge, but it would be more convincing to show that FedMuon can indeed converge under the same problem setting.
3. While illustrating the bias issue with a divergence example is interesting, similar bias phenomena commonly appear in other contexts such as bilevel optimization. A typical and simple approach in those settings is to communicate the momentum across clients in LocalMuon. However, the paper lacks discussion on why the authors chose not to adopt such straightforward momentum communication and instead opted for a control-variate-based correction mechanism.
4. There is no discussion of heterogeneous settings in the theoretical section. Given that the experiments include heterogeneous data distributions, the theoretical analysis should also address how FedMuon behaves under such non-IID conditions.
5. The empirical validation is confined to two small-scale vision datasets (FashionMNIST, CIFAR-10). The empirical validation should include large-scale or non-vision tasks to better demonstrate how FedMuon scales to realistic federated learning workloads or LLM fine-tuning scenarios.
6. Figure 2 provides an example of how the number of Newton–Schulz iterations $T$ affects performance. However, prior works such as [1] and [2] commonly fix $T=5$, and [3] shows that increasing beyond 5 yields no performance gain but adds unnecessary wall-clock time. Thus, the absence of $T\geq5$ in Figure 2 makes the empirical validation less convincing.

[1] Jordan et al., Muon: An optimizer for hidden layers in neural networks, 2024.

[2] Refael et al., Sumo: Subspace-aware moment-orthogonalization for accelerating memory-efficient llm training, 2025.

[3] Semenov et al., Benchmarking Optimizers for Large Language Model Pretraining, 2025.

**Questions:**

See weaknesses above.

---

> ### Author Response · Authors · 2025-11-28
>
> Thank you for your feedback. We have carefully addressed each point below, and we would be very grateful if these clarifications and revisions could be reflected in the final score. In particular, we respond to the concerns regarding the need for a clearer theoretical connection to the bias-mitigation mechanism and the convergence guarantee of FedMuon under the same setting in which LocalMuon stagnates, as well as the other issues raised by the reviewer.
>
> > The algorithm section (Section 4) only offers an intuitive, SCAFFOLD-like rationale for bias mitigation, but lacks formal theoretical support. It would be helpful to include a short proof sketch or lemma explicitly showing which terms or steps contribute to bias reduction.
>
> The formal theoretical support for FedMuon is given by Theorems 2 and 3, which prove convergence under both exact and inexact LMOs.
> We agree that it would be helpful to connect the intuition and the proofs more directly.
>
>
> > Appendix D includes an example to prove why LocalMuon does not converge, but it would be more convincing to show that FedMuon can indeed converge under the same problem setting.
>
> In Theorem 2 we provide convergence analysis in the non-convex case.
> The convex case is contained as a special instance.
> Hence, under the assumptions of Theorem 2, FedMuon converges to a stationary point even in the setting considered in Theorem 1, where LocalMuon stagnates.
> We will make this connection explicit in the main text.
>
> > While illustrating the bias issue with a divergence example is interesting, similar bias phenomena commonly appear in other contexts such as bilevel optimization. A typical and simple approach in those settings is to communicate the momentum across clients in LocalMuon. However, the paper lacks discussion on why the authors chose not to adopt such straightforward momentum communication and instead opted for a control-variate-based correction mechanism.
>
> Communicating momentum across clients is a natural baseline, but in the presence of a biased LMO and data heterogeneity it is difficult to control the combined error analytically.
> Control variates provide a framework that simultaneously corrects for heterogeneity and LMO bias and is compatible with our inexact LMO analysis for arbitrary Newton--Schulz steps.
> We will add a discussion that compares momentum communication and control variates and explains why we chose the latter design.
>
>
> > There is no discussion of heterogeneous settings in the theoretical section. Given that the experiments include heterogeneous data distributions, the theoretical analysis should also address how FedMuon behaves under such non-IID conditions.
>
> We respectfully disagree with this comment.
> Our analysis already covers heterogeneous data through standard assumptions that allow clients to have arbitrary dissimilar loss functions $f_i$, which are stated in Sec. 5.1.
> We will highlight more clearly in the main text that these assumptions correspond to non-IID data distributions across clients and that FedMuon is robust to data heterogeneity, as FedMuon can converge under arbitrary heterogeneous settings.
>
>
> > The empirical validation is confined to two small-scale vision datasets (FashionMNIST, CIFAR-10). The empirical validation should include large-scale or non-vision tasks to better demonstrate how FedMuon scales to realistic federated learning workloads or LLM fine-tuning scenarios.
>
>
> We agree that larger scale and non-vision tasks are very important.
> As mentioned above, we are extending our experiments to more realistic federated workloads, including transformer based models and NLP tasks, and will include representative results and discussion in a revised version.
>
>
> > Figure 2 provides an example of how the number of Newton–Schulz iterations
>  affects performance. However, prior works such as [1] and [2] commonly fix $T=5$, and [3] shows that increasing beyond 5 yields no performance gain but adds unnecessary wall-clock time. Thus, the absence of $T \geq 5$ in Figure 2 makes the empirical validation less convincing.
>
> Thank you for this comment.
> We added the results for $T=5$ to Figure 2.
> The observation is consistent with prior works: performance saturates after a small number of Newton-Schulz iterations, and increasing $T$ beyond $5$ does not improve performance, although our theory remains valid for any $T$.

---

### Official Review · Reviewer_paxH · 2025-10-26

**Soundness:** 3
**Presentation:** 2
**Contribution:** 3
**Rating:** 4
**Confidence:** 4

**Summary:**

This paper proposes FedMuon, a federated optimization algorithm that extends the recently proposed Muon optimizer—an LMO (Linear Minimization Oracle)-based method—into the federated learning setting. The authors identify that directly applying Muon in FedAvg (termed LocalMuon) fails to converge due to the bias of the LMO operator. To mitigate this, FedMuon introduces a bias-correction mechanism inspired by SCAFFOLD, along with a theoretical convergence analysis that also accounts for inexact LMO computations via the Newton–Schulz iteration. The authors provide both theoretical results and empirical validation on benchmark datasets (FashionMNIST, CIFAR-10) demonstrating FedMuon’s improved convergence and accuracy over existing methods such as FedAvg, FedAdam, and SCAFFOLD.

**Strengths:**

1.Novel and technically solid contribution.
The paper identifies a subtle but important issue (bias of the LMO in federated settings) and provides a theoretically justified correction method. The proofs (Theorem 1–3) are rigorous and clearly stated.

2.Theoretical generality.
The analysis handles both exact and inexact LMOs, showing convergence under any number of Newton–Schulz iterations—a strong and non-trivial result rarely seen in this line of work.

3.Experimental validation.
The experiments convincingly show FedMuon outperforming strong baselines under both homogeneous and heterogeneous data distributions. The empirical trends match the theoretical predictions (faster convergence as LMO accuracy increases).

4.Bridging two active research areas.
The paper effectively connects recent LMO-based optimization (Muon) with federated learning theory, which is timely and relevant for distributed LLM optimization

**Weaknesses:**

1.Limited experimental diversity.
The experiments are conducted only on small-scale benchmarks (FashionMNIST and CIFAR-10). Demonstrating results on larger or more realistic FL setups (e.g., cross-device or NLP tasks) would strengthen the empirical claims. ViT model or roberta-base model，imagenet,CIFAR100 data.

2.The proposed method simply combines the MUON and SCAFFOLD algorithms, and its communication cost is about twice that of the original approach.

3.The experimental results do not show a significant advantage.

4.The theoretical analysis of the paper is solid, but there is still room for improvement in the writing and experimental sections.

**Questions:**

1. Can the communication cost be further reduced?

2.Is the method still effective for training and fine-tuning large models?

3.Can the code be open-sourced?

4.The algorithm’s line 8 is written differently from the MUON optimizer — why is it designed this way?

5.Is the number of local iterations too small（K=5） ?

---

> ### Author Response · Authors · 2025-11-28
>
> > 1.Limited experimental diversity. The experiments are conducted only on small-scale benchmarks (FashionMNIST and CIFAR-10). Demonstrating results on larger or more realistic FL setups (e.g., cross-device or NLP tasks) would strengthen the empirical claims. ViT model or roberta-base model, imagenet,CIFAR100 data.
>
>
> We agree that larger and more realistic setups are very valuable.
> In this work we intentionally focus on controlled benchmarks that closely match the theoretical assumptions, to clearly demonstrate the phenomena studied in the analysis.
> We are extending our experiments to larger vision and NLP models, and we will include representative results in a revised version.
>
>
> > 2.The proposed method simply combines the MUON and SCAFFOLD algorithms, and its communication cost is about twice that of the original approach.
>
> FedMuon is designed specifically for the biased LMO setting in federated learning.
> The LMO is applied to a bias corrected momentum direction that couples client side and server side control variates, which is essential for our convergence proofs.
> SCAFFOLD appears only as a special case when the LMO and momentum are removed or tuned in a particular way.
> Regarding communication, FedMuon has essentially the same communication pattern per round as SCAFFOLD, since both transmit control variates; standard compression techniques can be applied on top of FedMuon and are orthogonal to our analysis.
>
>
> > 3.The experimental results do not show a significant advantage.
>
> We respectfully disagree.
> For instance, in the heterogeneous setting of Fig. 1b, FedMuon outperforms SCAFFOLD(Adam) by $3.0\\%$ and $10.3\\%$ on FashionMNIST and CIFAR-10, respectively, and converges in fewer communication rounds.
> These improvements are consistent across seeds and align with our theoretical predictions.
>
> > 1. Can the communication cost be further reduced?
>
> Yes.
> Communication compression techniques (such as quantization or sparsification) can be incorporated into FedMuon in the same way as for SCAFFOLD, without changing the core algorithmic ideas.
> A detailed study of compressed FedMuon is an interesting direction for future work.
>
> > 2.Is the method still effective for training and fine-tuning large models?
>
> This is an important question.
> We are currently running experiments with larger models (including transformer based architectures) and observe that the qualitative behavior is consistent with our small scale results.
> We will include these results and discussion of scalability in a revised version.
>
>
> > 3.Can the code be open-sourced?
>
>
> We have included our code in the supplementary material and plan to release a cleaned up version publicly.
>
> > 4.The algorithm’s line 8 is written differently from the MUON optimizer — why is it designed this way?
>
>
> The two momentum update rules are essentially equivalent.
> By slightly adjusting the scale of the step size one can recover the original Muon momentum update.
> We use the current form because it simplifies the analysis, in particular when combined with the control variates.
>
> > 5.Is the number of local iterations too small(K=5)?
>
> We follow common practice in federated learning and choose a small number of local steps to balance local computation and communication.
> In preliminary experiments with larger $K$ we observed the same qualitative behavior and a similar relative advantage of FedMuon over the baselines.
> We promise to add these results to the camera-ready version.

---

> > ### Comment · Reviewer_paxH · 2025-11-28
> >
> > Would the authors be able to include additional experiments to further address the issues mentioned above?

---

> > > ### Author Response · Authors · 2025-11-28
> > >
> > > We are conducting experiments using CIFAR-100 and Nano-GPT.
> > > We will include these results in the camera-ready version to address your concerns.

---

### Official Review · Reviewer_B6xb · 2025-10-29

**Soundness:** 3
**Presentation:** 2
**Contribution:** 3
**Rating:** 4
**Confidence:** 3

**Summary:**

This paper focuses on improving federated learning optimization by addressing the bias issues when applying the Muon optimizer, which uses Linear Minimization Oracle (LMO), in the FedAvg framework. The authors show that directly using Muon within FedAvg, referred to as LocalMuon, fails to converge due to the bias introduced by the LMO. To overcome this, they introduce FEDMUON, an optimization algorithm that incorporates a bias-correction mechanism inspired by SCAFFOLD, ensuring convergence. The theoretical analysis includes the convergence rate under inexact LMO computations, handled via Newton-Schulz iterations. Experiments on two benchmark datasets, FashionMNIST and CIFAR-10, demonstrate that FEDMUON outperforms several existing methods, including FedAvg, FedAvg (Adam), SCAFFOLD, and SCAFFOLD (Adam).

**Strengths:**

$\cdot$ The paper is well-motivated. Muon can outperform traditional adaptive optimizers like Adam in centralized settings, but its direct application to federated learning, specifically in FedAvg, fails to converge due to bias introduced by the LMO.

$\cdot$ The use of Newton-Schulz iterations to approximate the Linear Minimization Oracle (LMO) in FEDMUON is a key innovation, as it enables efficient bias correction while maintaining strong convergence guarantees even with inexact LMO solutions.

$\cdot$ The experimental results in the paper are highly consistent with the theoretical analysis, supporting the convergence guarantees of FEDMUON derived in the theory.

**Weaknesses:**

$\cdot$ It seems that the authors just incorporate the existing Muon to SCAFFOLD framework, which is just incremental. The authors should clarify their original contribution.

$\cdot$ The experiments are conducted on small datasets and simple models. Since optimizers like Adam perform better with larger models, comparisons on more complex models like Transformers would provide a better evaluation of FEDMUON's scalability and effectiveness.

$\cdot$ It would be better to report the time cost or communication cost of FEDMUON and baseline methods to empirically demonstrate that FEDMUON's advantages hold in terms of efficiency as well as performance.

**Questions:**

$\cdot$ Can FEDMUON be effective on larger models, such as transformers, and not just small-scale datasets and simple models?

$\cdot$ Besides SCAFFOLD from 2020, can the authors compare FEDMUON with more advanced federated learning algorithms？

$\cdot$ Could the authors report the communication or time costs of FEDMUON and baseline methods to demonstrate its efficiency advantages?

---

> ### Author Response · Authors · 2025-11-28
>
> Thank you for the reviewer’s comments. Below, we clarify our contributions. In particular, our paper carefully derived the lower bound of LocalMuon and provided the analysis of (Fed)Muon with the inexact LMO. We hope that these clarifications and revisions can be taken into account when determining the final score.
>
>
> > It seems that the authors just incorporate the existing Muon to SCAFFOLD framework, which is just incremental. The authors should clarify their original contribution.
>
> Our main contributions are:
> (i) a new lower bound that formally shows the failure of LocalMuon under heterogeneous data,
> (ii) the design of FedMuon, which uses a bias corrected momentum direction inside the LMO and strictly generalizes SCAFFOLD, and
> (iii) the first convergence analysis of a Muon based federated optimizer with inexact LMO for any number of Newton--Schulz iterations.
> These results are, to the best of our knowledge, new in both the Muon and federated learning literature.
> We would appreciate it if the reviewer also considers these novelties in our convergence analysis.
>
> >  The experiments are conducted on small datasets and simple models. Since optimizers like Adam perform better with larger models, comparisons on more complex models like Transformers would provide a better evaluation of FEDMUON's scalability and effectiveness.
>
> We agree that larger models are important.
> In this first work we intentionally start from smaller models that closely match the assumptions used in our analysis, to isolate the effect of LMO bias and inexactness.
> We are already running experiments with transformer architectures and will include representative results and a discussion of scalability in a revised version.
>
>
> >  It would be better to report the time cost or communication cost of FEDMUON and baseline methods to empirically demonstrate that FEDMUON's advantages hold in terms of efficiency as well as performance.
>
> FedMuon and SCAFFOLD have the same communication pattern per round, since both exchange control variates in addition to model parameters.
> For computational efficiency, we promise to report the training time in the camera-ready version.
>
> > Besides SCAFFOLD from 2020, can the authors compare FEDMUON with more advanced federated learning algorithms?
>
> Our experimental setting is designed to isolate the role of the biased LMO and to compare with methods that differ mainly in the optimizer (SGD, Adam, Muon).
> In future work we plan to add comparisons with more recent federated algorithms.
> We expect FedMuon to be complementary to these methods, since it focuses on the optimizer rather than personalization or architecture level changes.
>
>
> > Could the authors report the communication or time costs of FEDMUON and baseline methods to demonstrate its efficiency advantages?
>
> We promise to add these results to the camera-ready version.

---

### Official Review · Reviewer_E1uB · 2025-11-11

**Soundness:** 1
**Presentation:** 2
**Contribution:** 1
**Rating:** 2
**Confidence:** 5

**Summary:**

This paper proposes FedMuon, an algorithm improving usage of Muon in federated learning. They propose Scaffold-like modification in Muon's momentum term before LMO and observe improvement in stability. They also show some theoretical analysis in spectrum norm.

**Strengths:**

The writing is clear and easy to follow.

**Weaknesses:**

I'm not convinced by the paper's solidity and novelty.
1. The analysis in Section 3 regarding Muon’s failure to converge in FL seems rather trivial. Similar reasoning could apply to any momentum-based optimization algorithm. This claim should be supported by experimental evidence. In fact, based on the only experiments on LocalMuon shown in Figure 1, I don’t observe significant convergence issues. While there are some oscillations, I believe standard techniques such as learning rate or hyperparameter tuning could easily address them.

2. The modification is just including Scaffold-like updates in Muon, which seems quite limited.

3. The theoretical analysis is standard by combing Muon and Scaffold and classical SGD together.

4. It’s already 2025, yet the experiments are limited to FashionMNIST and CIFAR-10 under very restricted settings, with only two baselines (FedAvg and SCAFFOLD). This level of evaluation would have been insufficient even several years ago. Moreover, the final accuracy shows almost no improvement over the baselines.

**Questions:**

What weight decay term was used in the experiments? This factor has a significant impact on the results, and certain related techniques (e.g., [1]) can help stabilize training.

[1] FedNAR: Federated Optimization with Normalized Annealing Regularization

---

> ### Author Response · Authors · 2025-11-28
>
> Thank you for the feecback.
> Below, we addressed your concerns. We would like to emphasize that our paper provided a novel lower bound for LMO-based FedAvg (LocalMuon), and provided a novel analysis of FedMuon with inexact LMO, which is not marginal. See also the response in "General Response". We kindly ask the reviewer to reconsider the evaluation, and if you reflect our response to the final score, we would appreciate it.
>
> > [...] In fact, based on the only experiments on LocalMuon shown in Figure 1, I don’t observe significant convergence issues. While there are some oscillations, I believe standard techniques such as learning rate or hyperparameter tuning could easily address them.
>
> We disagree with this comment.
> The hyperparameters were carefully tuned as discussed in Sec. G.
> Experiments show that LocalMuon behaves reasonably in the homogeneous setting, but fails in the heterogeneous setting, where it stagnates in accordance with our lower bound.
> This behavior is not removed by hyperparameter tuning and reflects a structural limitation that comes from the biased LMO in the federated setting.
>
>
> > The modification is just including Scaffold-like updates in Muon, which seems quite limited.
>
> FedMuon is not obtained by directly inserting Muon into SCAFFOLD.
> The LMO in FedMuon is applied to a bias-corrected momentum that couples local momentum with both client and server control variates, and this specific structure is required for our convergence proof, in particular for the inexact LMO analysis, to go through.
> If we remove the LMO and choose parameters in a particular way, SCAFFOLD is recovered as a special case, which shows that FedMuon strictly generalizes SCAFFOLD rather than reducing to it.
>
> Please note that we also provide the analysis with \emph{an inexact solution of LMO}.
> This is the first analysis that shows that (Fed)Muon can converge for any number of Newton–Schulz iterations.
> We would appreciate it if the reviewer also considers this novelty in our convergence analysis.
>
>
> > The theoretical analysis is standard by combing Muon and Scaffold and classical SGD together.
>
> Our analysis goes beyond a direct combination.
> In particular, we provide, to the best of our knowledge, the first convergence proof for a federated Muon type method with inexact LMO, valid for any number of Newton--Schulz iterations.
> The dependence of the rate on the inexactness of the LMO and on the control variates requires a dedicated treatment that is specific to this setting.
>
> > It’s already 2025, yet the experiments are limited to FashionMNIST and CIFAR-10 under very restricted settings, with only two baselines (FedAvg and SCAFFOLD). This level of evaluation would have been insufficient even several years ago. Moreover, the final accuracy shows almost no improvement over the baselines.
>
> Our primary goal is to understand how Muon and LMO based optimization behave in federated learning with client heterogeneity and inexact LMOs.
> For this reason we focus on controlled benchmarks that match the theory and include FedAvg, FedAvg(Adam), SCAFFOLD, and SCAFFOLD(Adam) as baselines.
> In the heterogeneous setting FedMuon improves both convergence speed and final accuracy over these methods; for instance, in Fig.~1b FedMuon achieves several percentage points higher accuracy than SCAFFOLD(Adam).
> We agree that larger models and tasks are valuable and we are running additional experiments with transformer based models.

---

### Author Response · Authors · 2025-11-28
**General Response**

We thank all reviewers for their careful reading and constructive comments.
Several points in the reviews currently describe our work as essentially applying Muon inside SCAFFOLD, with limited novelty.
We would like to clarify that our contribution is more fundamental and has three main parts:


* **New lower bound for LMO based FedAvg (LocalMuon).** We prove that using an LMO based optimizer as the local method in FedAvg (LocalMuon) can stagnate under heterogeneous data, even on convex problems. In conctrast to Local SGD, whose convergence just becomes slower under the heterogeneous setting, LocalMuon does not converge. This is a consequence of the biased LMO in the federated setting and, to the best of our knowledge, has not been shown before.
* **New algorithm FedMuon that does not reduce to SCAFFOLD.** FedMuon applies the LMO to a carefully constructed bias corrected momentum that couples local momentum with both client and server control variates. SCAFFOLD is recovered only as a special case when the LMO and momentum are removed or chosen in a particular way, so FedMuon strictly generalizes SCAFFOLD rather than being identical to it.
* **First convergence analysis for federated Muon with inexact LMO and arbitrary Newton-Schulz steps.** Our theory shows that FedMuon converges for any number of Newton-Schulz iterations and quantifies how the rate depends on the inexactness of the LMO. Existing analyses of Muon either assume an exact LMO or fix the number of Newton-Schulz iterations to some specific value and do not provide guarantees for arbitrary choices in a heterogeneous federated setting.

Muon is already an important optimizer in large scale training, yet before this work there was no federated learning method with convergence guarantees that uses Muon, or more generally LMO based optimization, under client heterogeneity.
FedMuon closes this gap with both a new algorithm and new theory.
Our experiments are designed to match the theoretical setting and to isolate the role of LMO bias and inexactness; we agree that larger models and benchmarks are valuable and we are already extending our experiments in this direction.

---

### Meta-Review · Area_Chair_fayf · 2026-01-08

**Summary:**

This paper studied the popular Muon optimizer for federated learning, and proposed a matrix-variate  federated learning (FedMuon) algorithm based on the existing SCAFFOLD method. Theoretically, it first proved that using an LMO based optimizer as the local method in FedAvg (LocalMuon) can stagnate under heterogeneous data, even on convex problems. Moreover, it provided the convergence analysis for the proposed FedMuon algorithm, and proved that the FedMuon converges for any number of Newton-Schulz iterations and quantifies how the rate depends on the inexactness of the LMO. It also provided some numerical experiments to demonstrate efficiency of the proposed algorithm.

Compared with the most related papers [1,2], this paper has two novelties as follows:

1) This paper first proved that using an LMO based optimizer as the local method in FedAvg (LocalMuon) can stagnate under heterogeneous data, even on convex problems.

2) This paper also first proved that the FedMuon converges for any number of Newton-Schulz iterations and quantifies how the rate depends on the inexactness of the LMO.

So I recommend accepting this paper.  I suggest the authors follow the reviewers' suggestions during the camera-ready preparation,
especially in experimental aspects.


[1] FedMuon: Accelerating Federated Learning with Matrix Orthogonalization  (https://arxiv.org/pdf/2510.27403)

[2] On Provable Benefits of Muon in Federated Learning (https://arxiv.org/pdf/2510.03866?)

**Reviewer Concerns:**

The authors basically have addressed all reviewers' concerns in the rebuttal.

**Reviewer Scores:**

I have checked that Reviewer paxH and Reviewer YvBN are the authors of the papers [1,2], which are the similar as this paper.
Considering the conflict of interest, so the importance of scores is reduced.

[1] FedMuon: Accelerating Federated Learning with Matrix Orthogonalization  (https://arxiv.org/pdf/2510.27403)

[2] On Provable Benefits of Muon in Federated Learning (https://arxiv.org/pdf/2510.03866?)

---

### Decision · Program_Chairs · 2026-01-26

Accept (Poster)